# STRATEGIC CLASSIFICATION WITH EXTERNALITIES

[*]**Safwan Hossain**[1], [*]**Evi Micha**[2], **Yiling Chen**[1], **Ariel Procaccia**[1]

[1] Harvard University
[2] University of Southern California
`shossain@g.harvard.edu, evi.micha@usc.edu`

## ABSTRACT

We propose a new variant of the strategic classification problem: a principal reveals a classifier, and $n$ agents report their (possibly manipulated) features to be classified. Motivated by real-world applications, our model crucially allows the manipulation of one agent to affect another; that is, it explicitly captures inter-agent externalities. The principal-agent interactions are formally modeled as a Stackelberg game, with the resulting agent manipulation dynamics captured as a simultaneous game. We show that under certain assumptions, the pure Nash Equilibrium of this agent manipulation game is unique and can be efficiently computed. Leveraging this result, PAC learning guarantees are established for the learner: informally, we show that it is possible to learn classifiers that minimize loss on the distribution, even when a random number of agents are manipulating their way to a pure Nash Equilibrium. We also comment on the optimization of such classifiers through gradient-based approaches. This work sets the theoretical foundations for a more realistic analysis of classifiers that are robust against multiple strategic actors interacting in a common environment.

## 1 INTRODUCTION

Machine learning algorithms are increasingly deployed in high-stakes decision making, including loan applications, school admissions, hiring, and insurance claims (Bejarano Carbo, 2021; Harwell, 2022; Kumar et al., 2022). Relying on past data as a reference, these algorithms use features of a candidate to determine their merit for the given task. The interactions in these settings, however, often involve *strategic agents* who may manipulate their features if doing so yields a more favorable outcome. This presents a significant challenge to the algorithm if it is not trained to anticipate such behavior since the training and test distributions no longer match. Correspondingly, a large and growing literature on *strategic classification* has emerged to understand the dynamics of this behavior and propose strategies for learning in such settings (Hardt et al., 2016).

By and large, the existing literature roughly models the interaction as follows: The learner (or principal) deploys a classifier, and an agent with feature $x$ observes this and may choose to report a manipulated feature $x'$ to obtain a better outcome. Manipulation is not considered free since it may involve some additional effort or risk. While the classifier may be deployed for any number of agents, agent interactions with the learner are crucially assumed to be independent; in other words, one agent's action has no influence on another. We posit that this ignores a critical aspect of multi-agent interactions in a shared environment: agents' actions exert *externality* on one another. This is not a new observation: indeed, the notion that agents in a system are affected by a common resource has been a core component of economic models for over a century (Pigou, 1920; Coase, 1960). It has not, however, been formally studied in the strategic classification setting despite its relevance and applicability.

Consider the example Hardt et al. (2016) used to initiate this literature: the number of books in the parent's household may be a valid feature for university admissions given its correlation with student success (Evans et al., 2010). Since "books are relatively cheap", this allows parents to game the system by buying "an attic full of unread books"(Hardt et al., 2016). It is immediate, however, that if all parents do this, then the price of books no longer stays cheap. This is the externality induced

---

[*]Corresponding Authors; Equal Contribution

by everyone's actions. Externality, however, can not only model demand for lucrative features but also the additional risk or burden associated with manipulation. When a university is deciding which students to accept from a high school using a classification algorithm, applicants may report features such as class rank, GPA, volunteering affiliations, and so on. Naturally, only a handful of students can be at the top of their class or be affiliated with a specific organization. If only a handful of students manipulate their features, it may not be statistically discernible. However, if many claim to be at the top of their class or work with the same organization, this can stand out and lead the university to audit the applications further. Similarly, a few applicants inflating their credit score on a loan application may be treated as an anomaly; a large pool of applicants doing so becomes a systematic problem that demands a reexamination of the loan process, causing a burden on all.

The preceding examples illustrate that the demand for books, the risk faced by a student, or the burden on the borrower is not solely determined by an individual's action but also influenced by the interactions of others within the same system. This naturally affects how, what, and if an agent manipulates. This phenomenon applies to most settings where strategic classification is pertinent. As such, it is imperative to correctly model and understand this phenomenon to ensure that classifiers deployed in sensitive settings act as intended. We specifically study the following questions:

- How should we jointly model the strategic interactions between agents alongside those with the learner?
- What is the appropriate notion of equilibrium in this setting and what properties does it have?
- What learning guarantees can be given for classifiers in such settings?

## 1.1 Our Contributions

We study the problem of deploying classifiers in strategic multi-agent settings with inter-agent externalities from a theoretical perspective. Consistent with prior literature, the interaction between the learner and the agents is modeled as a Stackelberg game: the learner first commits to a classifier to which agents can respond. The resulting inter-agent interactions are captured as a simultaneous game, and the *Stackelberg-Nash Equilibrium* is proposed as the solution concept for all interactions. We precisely define these aspects of the *multi-agent strategic classification game* in Section 2. Learning in this setting is challenging, not least due to the possible multiplicity of equilibrium, their computation, and their dynamics due to a changing classifier. Section 3 motivates a set of structural assumptions on the cost and externality, under which the inter-agent simultaneous game has a unique pure Nash Equilibrium that is efficiently computable. Building on this insight, Section 4 comments on the regularity of the Nash Equilibrium under changing classifiers, and provides probably approximately correct (PAC) learning guarantees for computing the Stackelberg-Nash Equilibrium. Intuitively, it is possible to learn loss-minimizing classifiers that generalize to a random number of agents manipulating to achieve a Nash Equilibrium. We also differentiate through the equilibrium solution to explicitly characterize the loss gradient, illustrating the feasibility of gradient-based optimization algorithms. Section 5 uses our motivating examples to model some externality functions that can be captured by our general framework. Lastly, in Section 7, we discuss some limitations of our work alongside technical and conceptual extensions.

## 1.2 Related Work

Our work builds on the growing literature on strategic classification (Hardt et al., 2016; Brückner et al., 2012; Brückner & Scheffer, 2009). In the basic setting, a learner seeks to release a classifier that accounts for the fact that strategic agents may misreport their true features to maximize their utility, which is determined by the likelihood of a positive outcome and the cost incurred for misreporting.

Recent papers have studied extensions of the basic strategic classification setting. For example, Levanon & Rosenfeld (2022) introduce a setting where agents' utility functions capture different intentions beyond simply maximizing for the positive outcome. Others aim to account for limited information (Ghalme et al., 2021; Harris et al., 2022; Bechavod et al., 2022), unknown utilities (Dong et al., 2018) and causal effects (Miller et al., 2020; Horowitz & Rosenfeld, 2023). These extensions do not handle externalities or multi-agent behaviour in general.

While motivated by a different aspect of strategic classification, the work of Eilat et al. (2023), which studies strategic classification in a graph setting, has conceptual similarities with our work. Node classification on graphs naturally depends on neighboring nodes' features, which strategic agents can exploit. While this implicitly models agent interactions as not wholly independent, there are several fundamental differences from our work. First, agents in their model are interrelated due to the classifier outcome of one agent depending on neighboring nodes; in contrast, we consider the classifier outcome for an agent to only depend on their feature, with the externality explicitly capturing additional risk or cost to agents due to others also interacting in the system. This better aligns with our motivation to capture classification dynamics in competitive high-stakes settings. Furthermore, they consider agents myopically best responding over a sequence of rounds whereas we consider the performance of a classifier at a Nash Equilibrium.

Further afield, machine learning researchers have investigated the effect of strategic behavior on social welfare (Milli et al., 2019; Haghtalab et al., 2020; Kleinberg & Raghavan, 2020) or on different groups of agents (Hu et al., 2019). However, none of these papers consider the direct impact that manipulation by others can have on each individual agent. The impact of such externalities has, nonetheless, been studied in computational problems like auctions (Agarwal et al., 2020), data markets (Hossain & Chen, 2024), facility location (Li et al., 2019), and for long-term fairness accountability (Heidari et al., 2019).

## 2 MODEL

**Preliminaries:** For $k \in \mathbb{N}$, let $[k] = \{1, \dots, k\}$. We consider $k$ strategic agents interacting with a learner who releases a classifier parametrized by weights $\boldsymbol{\omega}$. Using one of our running examples, this corresponds to $k$ students applying for admission to a university, which decides to accept or reject using classifier $f_{\boldsymbol{\omega}}$. Let $\boldsymbol{x}_i \in \mathbb{R}^d$ and $y_i \in \{-1, +1\}$ denote the feature vector and true class of agent $i$ respectively, with $(\boldsymbol{x}_i, y_i) \sim \mathcal{D}$. Let $\boldsymbol{X} \in \mathbb{R}^{k \times d}$ denote the $k$ agents' feature matrix, and $\boldsymbol{y} \in \{-1, +1\}^k$ the vector of their true class labels[1], sampled independently from $\mathcal{D}$; we use the shorthand

$$(\boldsymbol{X}, \boldsymbol{y}) \sim \underbrace{\mathcal{D} \times \cdots \times \mathcal{D}}_{k} \triangleq \mathcal{D}^k.$$

We use the notation $\boldsymbol{X}_{i:}$ and $\boldsymbol{X}_{:j}$ to respectively refer to the $i^{th}$ row and $j^{th}$ column of the matrix $\boldsymbol{X}$, and $[\boldsymbol{X}_1; \boldsymbol{X}_2]$ to denote two matrices concatenated along rows.

Our focus will be on binary linear classifiers due to their wide-spread practical usage and popularity within the strategic classification literature[2] (Hardt et al., 2016; Dong et al., 2018; Bechavod et al., 2022); that is, $\boldsymbol{\omega} \in \Omega \subseteq \mathbb{R}^d$ and $f_{\boldsymbol{\omega}}(\boldsymbol{x}) = \langle \boldsymbol{x}, \boldsymbol{\omega} \rangle$ denotes the score for positive classification. The learner may use any Lipschitz loss function[3] to compute the loss and maximize the accuracy of their deployed classifier with respect to the true labels.

**Utility and Externality:** Following the standard strategic classification model, we consider a Stackelberg interaction between the learner and the agents (Hardt et al., 2016). That is, the learner first releases a classifier $f_{\boldsymbol{\omega}}$, and thereafter, agents submit their features to be classified. An agent need not be truthful and may instead submit a manipulated feature vector $\boldsymbol{x}_i'$ to receive a higher score for the positive class. Consistent with the strategic classification literature (Dong et al., 2018; Bechavod et al., 2022), we assume agent utilities are proportional to the score: $f_{\boldsymbol{\omega}}(\boldsymbol{x}_i')g^+(\boldsymbol{x}_i)$, where $g^+(\boldsymbol{x}_i) : \mathbb{R}^d \to \mathbb{R}$ is an agent specific gain for positive classification[4].

In line with prior work, we assume agent features lie within a bounded region (without loss of generality $[0, 1]^d$) and they incur a *cost* $c(\boldsymbol{x}_i, \boldsymbol{x}_i')$ in modifying their true feature vector. However, in contrast to these earlier models, we also consider agents' decisions causing externality to others. Conceptually, externality is the impact agents' actions have on one another when interacting within a

---

[1]In general, we use bold capital letters to refer to matrices, and bold lower case letters to refer to vectors.

[2]Our results can be generalized to multi-class linear classifiers.

[3]This includes nearly any reasonable loss function including Hinge Loss, Cross Entropy Loss, MSE, etc.

[4]It may be common to associate a type vector $\theta_i$ with an agent and express gain in terms of $g^+(\theta_i, \boldsymbol{x}_i)$. This is equivalent to our utility model as the feature vector can include the type. This type of dependence can be extended to our cost and externality functions as well.

common system. Formally, the negative externality suffered by agent $i$ due to an agent $j$ is captured by the function $t(\boldsymbol{x}_i, \boldsymbol{x}_i', \boldsymbol{x}_j, \boldsymbol{x}_j')$. We use $T(\boldsymbol{x}_i, \boldsymbol{x}_i', \boldsymbol{X}, \boldsymbol{X}') = \sum_{j \neq i} t(\boldsymbol{x}_i, \boldsymbol{x}_i', \boldsymbol{x}_j, \boldsymbol{x}_j')$ to denote the total externality suffered by agent $i$ due to all other participating agents' decision. In summary, the total utility of an agent $i$ in reporting $\boldsymbol{x}_i'$ in our multi-agent strategic classification game is given by:

$$ u_i(\boldsymbol{X}, \boldsymbol{X}', f_{\boldsymbol{\omega}}) = f_{\boldsymbol{\omega}}(\boldsymbol{x}_i') g^+(\boldsymbol{x}_i) - c(\boldsymbol{x}_i, \boldsymbol{x}_i') - T(\boldsymbol{x}_i, \boldsymbol{x}_i', \boldsymbol{X}, \boldsymbol{X}') \tag{1} $$

The standard literature on strategic classification assumes the principal knows the cost function (Hardt et al., 2016; Milli et al., 2019; Levanon & Rosenfeld, 2021). We extend this assumption to externality functions as well.

**Game-Theoretic Model:** Since an agent's total utility depends on others' reports, the interaction between the agents is naturally modeled as a game. Taking inspiration from prior work on strategic regression (Chen et al., 2018; Hossain & Shah, 2021) and related settings (Nakamura, 2015), we model this inter-agent interaction as a simultaneous game. Correspondingly, the *best response* of agent $i$ given classifier $f_{\omega}$ and reports of others is $\arg\max_{\boldsymbol{x}_i' \in [0,1]^d} u_i(\boldsymbol{X}, \boldsymbol{X}', f_{\boldsymbol{\omega}})$.

For a fixed classifier $f_{\boldsymbol{\omega}}$, the natural solution concept for the agent game is a *pure Nash Equilibrium* (PNE). Formally, a set of reported values $\boldsymbol{X}^q = [\boldsymbol{x}_i^q; \ldots; \boldsymbol{x}_i^q]$ is a PNE if for each agent $i$ and fixing the reported values of others, $\boldsymbol{x}_i^q$ is a best response for agent $i$. Intuitively, no agent can unilaterally improve their total utility at this equilibrium. Since PNE is not necessarily unique, let $\text{NE}(\boldsymbol{X}, f_{\boldsymbol{\omega}})$ denote the correspondence from the true features to the set of features reported at an equilibrium under classifier $f_{\boldsymbol{\omega}}$. That is: $\boldsymbol{X}^q \in \text{NE}(\boldsymbol{X}, f_{\boldsymbol{\omega}})$.

**Learning Problem:** While the relationship between the $k$ agents corresponds to a simultaneous game, the principal and agent interactions still induce a Stackelberg game since the principal first commits to a classifier $f_{\boldsymbol{\omega}}$. The Stackelberg equilibrium in the classical setting is with respect to a single agent best responding under the released classifier; however, our multi-agent setting implies the principal must now react according to the Nash Equilibrium induced by their chosen classifier. Thus, the principal's goal is to compute the *Stackelberg-Nash Equilibrium* (Xu et al., 2016), which corresponds to a classifier that performs optimally when sampled features reach a pure Nash Equilibrium with respect to the chosen classifier.

Since the number of participants in the classification setting (e.g., the number of students applying for admissions) is not fixed, let $k_{max}$ denote the maximum number of such participants, and let $\mathcal{K}$ denote a distribution over $[k_{max}]$. The principal has access to a dataset of $n$ labeled samples $\mathcal{S} = \{(\boldsymbol{X}_1, \boldsymbol{y}_1, k_1), \ldots, (\boldsymbol{X}_n, \boldsymbol{y}_n, k_n)\}$, where sample $j$ consists of $k_j \sim \mathcal{K}$ and $(\boldsymbol{X}_j, \boldsymbol{y}_j) \sim \mathcal{D}^{k_j}$. That is, the $j^{th}$ samples consist of the features and labels of the $k_j$ participating agents. The principal uses this training set $\mathcal{S}$ to learn a classifier that performs well in practice. The optimal classifier is formally defined as follows:

**Definition 1.** *The optimal classifier parameter $\boldsymbol{\omega}^*$ in the multi-agent strategic classification game with loss function $\ell$, participant distribution $\mathcal{K}$, and data distribution $\mathcal{D}$ is given by:*

$$ \boldsymbol{\omega}^* = \arg\min_{\boldsymbol{\omega} \in \Omega} \mathbb{E}_{k \sim \mathcal{K}} \mathbb{E}_{(\boldsymbol{X}, \boldsymbol{y}) \sim \mathcal{D}^k} \max_{\boldsymbol{X}^q \in \text{NE}(\boldsymbol{X}, f_{\boldsymbol{\omega}})} \left[ \sum_{i=1}^{k} \ell(y_i, f_{\boldsymbol{\omega}}(\boldsymbol{x}_i^q)) \right]. $$

The learning problem in this Stackelberg-Nash game is non-trivial for several reasons. First, it requires optimizing over the PNE of the sampled feature vectors. We discuss the efficient computation of this equilibrium in Section 3 under structural assumptions on the cost and externality. Second, we must not only generalize with respect to the equilibrium of sampled agents but also when the number of such agents is random. We address this and the optimization of $f_{\boldsymbol{\omega}}$ from a dataset $\mathcal{S}$ in Section 4.

## 3 EQUILIBRIUM PROPERTIES

Core to our problem is computing the PNE achieved by the sampled agents since it informs the loss suffered by the principal under their chosen classifier. Thus, understanding equilibrium properties such as existence, uniqueness, and computation is crucial to any learning algorithm. However, such properties cannot be precisely stated without a structured model of the agent utilities, which in our game is largely predicated upon the cost and externality. We formally state our assumptions below:

1. The cost is given by the $\ell_2$ norm of the manipulation vector: $c(\boldsymbol{x}_i, \boldsymbol{x}_i) = \alpha ||\boldsymbol{x}_i - \boldsymbol{x}'_i||_2^2$

2. Externalities are pairwise symmetric: $\forall i, j, \; t(\boldsymbol{x}_i, \boldsymbol{x}'_i, \boldsymbol{x}_j, \boldsymbol{x}'_j) = t(\boldsymbol{x}_j, \boldsymbol{x}'_j, \boldsymbol{x}_i, \boldsymbol{x}'_i)$.

3. Total externality faced by any agent $i$, $T(\boldsymbol{x}_i, \boldsymbol{x}'_i, \boldsymbol{X}, \boldsymbol{X}')$ is smooth and convex in the manipulation variables $(\boldsymbol{x}'_1, \ldots, \boldsymbol{x}'_k)$.

Initial work on strategic classification, including Hardt et al. (2016), assumed the cost function to be separable, a somewhat restrictive assumption. More recent work models cost as the norm of the manipulation vector $\boldsymbol{x}' - \boldsymbol{x}$ as it satisfies natural axioms associated with being a metric (Levanon & Rosenfeld, 2022; Horowitz & Rosenfeld, 2023). The most common choice is the $\ell_2$ norm, which we adopt (our results do hold for a larger class of cost functions - see the end of this section).

Structural assumptions on externality models are routine in economic literature. In competitive settings, which describe many high-stakes classification tasks, symmetric externalities are natural and commonplace (Goeree et al., 2005; Hossain & Chen, 2024). Settings such as data markets (Agarwal et al., 2020) and facility location (Li et al., 2019) further capture externality under a linear model, which our model subsumes. To give an example of a non-linear but convex externality, consider the loan application setting where a bank (the learner) is screening candidates for approval. The more candidates manipulate their reports, the more likely the bank's approval process will become stricter, adversely affecting all candidates. The function $t(\boldsymbol{x}_i, \boldsymbol{x}'_i, \boldsymbol{x}_j, \boldsymbol{x}'_j) = (||\boldsymbol{x}'_i - \boldsymbol{x}_i||_2 + ||\boldsymbol{x}'_j - \boldsymbol{x}_j||_2)^2$, which increases as the manipulation increases can model this setting. It is convex since the square of a non-negative convex function is convex. While this is one example, Section 5 contains a more detailed exploration of externality models, and shows that our results hold for a broader class of functions, including non-convex ones.

Returning to the PNE analysis of the manipulation game, we first show in Theorem 1 (proof in Appendix B) that when externalities are pairwise symmetric, the resulting interactions can be characterized by a potential game[5]. Such games use a single function, the potential function $\Phi$, to encapsulate the difference in utility to any agent $i$ due to their unilateral change in action (Monderer & Shapley, 1996). We formally define this below for convenience.

**Definition 2.** *An $n$ player simultaneous game is a potential game if there exists a potential function* $\Phi : \mathcal{A}^n \to \mathbb{R}$ *mapping from joint action space to the reals such that for any agent $i$: $u_i(a_i, \boldsymbol{a}_{-i}) - u_i(a'_i, \boldsymbol{a}_{-i}) = \Phi(a_i, \boldsymbol{a}_{-i}) - \Phi(a'_i, \boldsymbol{a}_{-i})$, where $u_i$ is the $i^{th}$ agent's utility.*

**Theorem 1.** *For any classifier $f_{\boldsymbol{\omega}}$, if the externalities are pairwise symmetric (Condition 2), the agent interactions constitute a potential game with the potential function:*

$$\Phi(\boldsymbol{X}, \boldsymbol{X}', f_{\boldsymbol{\omega}}) = \sum_{i=1}^{k} f_{\boldsymbol{\omega}}(\boldsymbol{x}'_i) g^{+}(\boldsymbol{x}_i) - \sum_{i=1}^{k} c(\boldsymbol{x}_i, \boldsymbol{x}'_i) - \sum_{i=1}^{k} \sum_{j>i} t(\boldsymbol{x}_i, \boldsymbol{x}'_i, \boldsymbol{x}_j, \boldsymbol{x}'_j).$$

Since all player incentives map to the potential function, PNE strategies exactly correspond to local maxima of the potential function (Roughgarden, 2010).[6] Indeed, if an agent could improve their utility by modifying their action, the value of the potential function at this updated action also increases. This has several implications. First, finding an equilibrium is equivalent to finding such local maxima, and second, properties of local maxima of $\Phi$ directly inform the multiplicity/uniqueness of the equilibria. Lastly, suppose agents sequentially play their best response from any starting point. In that case, the potential function value is strictly increasing, giving a simple local algorithm for arriving at a PNE.

Our next result shows that the potential function is strictly concave under our set of assumptions. For such functions, any local optimum is also the global optimum, such an optimum is unique, and the PNE must be at this optimum (Neyman, 1997). Correspondingly, the Nash Equilibrium is unique and can be written as the outcome of a convex optimization problem[7]. We present this formally below (proof in the Appendix B):

**Theorem 2.** *Under Conditions 1, 2 and 3 the potential function specified in Theorem 1 is strictly concave. Further, the Nash Equilibrium is unique and is given by the function,* $\arg\max_{\boldsymbol{X}' \in [0,1]^{k \times d}} \Phi(\boldsymbol{X}, \boldsymbol{X}', f_{\boldsymbol{\omega}})$.

---

[5]Appendix B also shows a type of global externality can also be modeled as a potential game.

[6]In this setting, a local maxima refers to a point that is optimal with respect to changes in any one direction.

[7]Maximizing a concave objective under convex constraints, is a convex optimization problem.

This implies that $NE(\boldsymbol{X}, f_{\boldsymbol{\omega}})$ is now a function mapping true value to the unique PNE. This resolves the ambiguity of learning in the presence of multiple equilibria. Further, this function is the outcome of a parameterized convex optimization problem (in terms of the classifier $f_{\boldsymbol{\omega}}$).

The result above hinges on the strict concavity of the potential function. While our stated conditions imply this, we note that Conditions 1 and 3 can be replaced with a weaker one:

> 1'. $\sum_i c(\boldsymbol{x}_i, \boldsymbol{x}_i') + \sum_i \sum_{j>i} t(\boldsymbol{x}_i, \boldsymbol{x}_i', \boldsymbol{x}_j, \boldsymbol{x}_j')$, *is smooth and strictly convex*

We denote this as the *cumulative impact of manipulation* — total cost and the sum of all combinations of externality. Due to our pairwise symmetry condition, $\sum_i \sum_{j>i} t(\boldsymbol{x}_i, \boldsymbol{x}_i', \boldsymbol{x}_j, \boldsymbol{x}_j') = \frac{1}{2} \sum_{i=1}^n T(\boldsymbol{x}_i, \boldsymbol{x}_i', \boldsymbol{X}, \boldsymbol{X}')$; since convexity is preserved through summation, for any strictly convex cost (i.e. Condition 1, $\ell_2$ norm squared cost) and any externality where $T(\boldsymbol{x}_i, \boldsymbol{x}_i', \boldsymbol{X}, \boldsymbol{X}')$ is convex (Condition 3) the updated condition above is immediately satisfied. This condition however is more general and allows us to capture a larger class of possibly non-convex externality within our framework. We have a detailed discussion of some of these richer models in Section 5.

## 4 LEARNING AND GENERALIZATION

We now tackle the problem of learnability in our setting. Broadly speaking, the learner's goal is to ensure that a classifier trained on a finite dataset to anticipate agents misreporting to an equilibrium, performs well on the broader population distribution which exhibits similar behaviour. A unique aspect of our setting is the number of agents participating in a given instance need not be fixed. Taking university admissions as an example, the number of applicants in a given year is not constant but rather a random variable. Our generalization result should accommodate this variability in the number of players in each instance, alongside the equilibrium they reach.

Recall from Section 2 that $k_{max}$ denotes the largest number of simultaneous participants with $\mathcal{K}$, a distribution over $[k_{max}]$, and $\mathcal{D}$, the data distribution that we sample from, i.e., $(\boldsymbol{x}_i, y_i) \sim \mathcal{D}$. The learner has access to a training dataset of $n$ instances, where each instance involves a sample $k_i \sim \mathcal{K}$, and then $k_i$ independent samples from the distribution $\mathcal{D}$. Observe that the following is an equivalent sampling procedure: sample $k_i \sim \mathcal{K}$, draw $k_{max}$ independent samples from $\mathcal{D}$, and then only consider the first $k_i$ elements. We use this latter procedure for the generalization result; formally, we define a joint distribution $\mathfrak{D} = \underbrace{\mathcal{D} \times \cdots \times \mathcal{D}}_{k_{max}} \times \mathcal{K}$. The learner has access to $n$ samples from this distribution, representing their training set: $\mathcal{S} = \{(\boldsymbol{X}_1, \boldsymbol{y}_1, k_1), \ldots, (\boldsymbol{X}_n, \boldsymbol{y}_n, k_n)\}$, where $\boldsymbol{X}_i \in \mathbb{R}^{k_{max} \times d}$, $\boldsymbol{y}_i \in \mathbb{R}^{k_{max}}$, and $k_i \in [k_{max}]$. Note that each sample/instance in this dataset can be seen as holding the attributes of $k_i$ individuals.

For any chosen classifier $f_{\boldsymbol{\omega}}$, the participating agents reach a PNE and the classifier suffers a resulting loss based on this. In the preceding section, we show this equilibrium to be the solution of a convex optimization problem. To provide formal learning guarantees, however, it is important to understand how the outcome of this optimization (i.e. the PNE) behaves as the classifier changes. Indeed, if the PNE drastically changes due to small changes in $\boldsymbol{\omega}$, learning can be challenging.

Let $NE(\boldsymbol{X}, f_{\boldsymbol{\omega}}, k) : \mathbb{R}^{k_{max} \times d} \times \mathbb{R}^d \times [k_{max}] \to \mathbb{R}^{k_{max} \times d}$, where the first $k$ rows of the output correspond to the equilibrium solution when $k$ agents are participating. We prove in the following result that the potential function optimum (and thus the PNE strategy) is Lipschitz continuous in $\boldsymbol{\omega}$. The theorem statement treats inputs to $\Phi$ and NE as vectors. This is without loss of generality since our $\boldsymbol{X} \in \mathbb{R}^{k_{max} \times d}$ feature matrix is equivalent to a $\boldsymbol{x} \in \mathbb{R}^{d k_{max}}$ vector, with the vector $\ell_2$ norm $||\boldsymbol{x}||_2$ equivalent to the matrix Frobenius norm $||\boldsymbol{X}||_F$.

**Lemma 1.** *Let $\Phi(\boldsymbol{x}, \boldsymbol{x}', f_{\boldsymbol{\omega}})$ be the potential function for $k_{max}$ agents ($\boldsymbol{x}, \boldsymbol{x}' \in \mathbb{R}^{d k_{max}}$). Then for $\eta = \frac{\gamma}{c}$, where $c = \frac{1}{2} \min_{\boldsymbol{x}', \boldsymbol{\omega}} (\lambda_{min}(\nabla^2_{\boldsymbol{x}'} \Phi(\boldsymbol{x}, \boldsymbol{x}', f_{\boldsymbol{\omega}})))$ and $\gamma = \max_{\boldsymbol{x}', \boldsymbol{\omega}} ||\nabla^2_{\boldsymbol{\omega} \boldsymbol{x}'} \Phi(\boldsymbol{x}, \boldsymbol{x}', f_{\boldsymbol{\omega}})|| + 1$, the function $NE(\boldsymbol{x}, f_{\boldsymbol{\omega}}, k)$ is $\eta$-Lipschitz in $\boldsymbol{\omega}$ (under the $|| \cdot ||_2$ norm) for any $k$.*

The proof (deferred to the Appendix C) is technical and in fact shows the Lipschitz continuity for a general class of parametrized convex optimization problems. Noting that local Lipschitzness suffices, we lower bound the difference between an optimal and sub-optimal solution using properties of strict concavity and smoothness of the potential function. The remainder of the proof leverages the continuity of the objective in both $\boldsymbol{x}'$ and $\boldsymbol{\omega}$ along with the insights above to establish

the Lipschitz-ness of the maximizer of $\Phi$. To relate this to $\text{NE}(\boldsymbol{X}, f_{\omega}, k)$, we define the optimization objective as summing the potential function over only the first $k$ participants with a trivial function involving the remaining agents.

Next, for a loss function $\ell(f_{\omega}(\boldsymbol{x}_i), y_i)$ defined on classifying an individual's reported features, the average loss on a sample consisting of $k$ participating strategic agents is given by $L(\boldsymbol{X}, \boldsymbol{y}, f_{\omega}, k) = \frac{1}{k} \sum_{i=1}^{k} \ell(f_{\omega}(\text{NE}(\boldsymbol{X}, f_{\omega}, k)_{i:}), y_i)$[8]. We now define the standard learning theory notions using this:

**Definition 3.** *For a classifier $f_{\omega}$, the* true risk $R(f_{\omega})$ *on distribution $\mathfrak{D}$ and the* empirical risk $\hat{R}(f_{\omega})$ *on a dataset $\mathcal{S}$ consisting of $n$ independent samples from $\mathfrak{D}$ is given respectively by:*

$$R(f_{\omega}) = \mathbb{E}_{(\boldsymbol{X}, \boldsymbol{y}, k) \sim \mathfrak{D}} L(\boldsymbol{X}, \boldsymbol{y}, f_{\omega}, k) \quad and \quad \hat{R}(f_{\omega}) = \frac{1}{n} \sum_{i=1}^{n} L(\boldsymbol{X}_i, \boldsymbol{y}_i, f_{\omega}, k_i).$$

We want to show that a classifier chosen to minimize the risk on a finite dataset $\mathcal{S}$, denoted by $\hat{f}_{\omega}$, will be close in a PAC sense, to the true risk minimizer of the population, denoted by $f_{\omega}^*$. Recall that we focus on linear classifiers and as such, our function class is the set of norm-constrained linear functions: $\{\langle \boldsymbol{\omega}, \boldsymbol{x}_i \rangle \text{ s.t. } ||\boldsymbol{\omega}|| \leq r\}$. We denote $\Omega = \{\boldsymbol{\omega} \text{ s.t. } ||\boldsymbol{\omega}||_2 \leq r\}$ as the corresponding parameter space. We now give our generalization guarantee:

**Theorem 3.** *For any $\lambda$-Lipschitz loss function $\ell$, linear function class $\Omega = \{\boldsymbol{\omega} \text{ s.t. } ||\boldsymbol{\omega}||_2 \leq r\}$, agent dynamics captured by a strictly concave potential function $\Phi(\boldsymbol{x}, \boldsymbol{x}', f_{\omega})$, and $\varepsilon > 0$, $\delta > 0$:*

$$n \geq \frac{8}{\varepsilon^2} \ln\left(\frac{e}{\gamma}\right) + d \ln\left(\frac{16(\lambda(d + \eta r)\gamma)}{\varepsilon}\right) \implies \mathbb{P}\left(|R(\hat{f}_{\omega}) - R(f_{\omega}^*)| \geq \varepsilon\right) \leq \delta$$

*where $\eta$ is the Lipschitz constant from Lemma 1.*

The proof (deferred to the Appendix C), notes that the average loss on the sample $(\boldsymbol{X}, \boldsymbol{y}, k)$ is a scalar regardless of the randomness of the number of participants. Further, due to the Lipschitz-ness of the NE function, the loss changes predictably due to a changing classifier. This allows us to apply a covering argument and bound the discretization error. While the sample complexity itself does not directly depend on $k_{max}$ (due mainly to the sample loss being scaled by the number of participants $\frac{1}{k}$) it does depend on the Lipschitz constant $\eta$. This constant is defined with respect to the potential function over the maximum number of $k_{max}$ participants (see Lemma 1). Sample complexity, thus, has an implicit dependence on $k_{max}$.

### 4.1 MINIMIZING EMPIRICAL RISK

While we have shown PAC learning is possible in our multi-agent strategic classification game, a cornerstone of learning is minimizing the empirical risk; that is, finding a classifier that minimizes loss incurred on the training set $\mathcal{S}$. Even for a convex loss function $\ell$ and linear classifier $f_{\omega}$, minimizing this across the samples of $\mathcal{S}$ is non-trivial since the NE function is the solution to an optimization problem. In general, there is no closed-form expression for the NE function and we cannot hope for this loss minimization problem to be convex.

Machine learning nonetheless has a vast literature on optimizing nonconvex loss functions; these, however, are largely gradient-based and require computing the loss gradient with respect to the learning parameter $\boldsymbol{\omega}$. This loss is computed with respect to the equilibrium outcome/score of a participating agent $i$: $f_{\omega}(\text{NE}(\boldsymbol{X}, f_{\omega})_{i:})$[9]. When $f_{\omega}$ is linear, this is simply $\langle \text{NE}(\boldsymbol{X}, f_{\omega})_{i:}, \boldsymbol{\omega} \rangle$. We now show when and how this gradient can be explicitly computed.

Let $v_i = \text{NE}(\boldsymbol{X}, f_{\omega})_{i:}$ and $z_i = f_{\omega}(\text{NE}(\boldsymbol{X}, f_{\omega})_{i:}) = f(v_i, \boldsymbol{\omega})$ for $i \in [k]$, noting that $z_i \in \mathbb{R}$ and $v_i \in \mathbb{R}^d$. The loss gradient is given as follows:

$$\nabla_{\boldsymbol{\omega}} L(\boldsymbol{x}, \boldsymbol{y}, f_{\omega}) = \frac{1}{k} \sum_{i=1}^{k} \nabla_{\boldsymbol{\omega}} \ell(f(\text{NE}(\boldsymbol{X}, f_{\omega})_{i:}, \boldsymbol{\omega}), y_i) = \frac{1}{k} \sum_{i=1}^{k} \nabla_{\boldsymbol{\omega}} \ell(z_i, y_i) = \frac{1}{k} \sum_{i=1}^{k} \frac{\partial \ell}{\partial z_i} \nabla_{\boldsymbol{\omega}} z_i$$

$$= \frac{1}{k} \sum_{i=1}^{k} \frac{\partial \ell}{\partial z_i} \left[ \nabla_{v_i}^T f \mathcal{J}_{\boldsymbol{\omega}}(\text{NE}(\boldsymbol{x}, f_{\omega})_{i:}) + \nabla_{\boldsymbol{\omega}} f \right]$$

---

[8]The notation $\text{NE}(\boldsymbol{X}, f_{\omega})_{i:}$ denotes the equilibrium report of the $i^{th}$ agent.

[9]Since we are concerned with the loss on a sample, we assume $k = k_{max}$ and drop the dependence on $k$.

We use the chain rule here, with $\mathcal{J}_{\boldsymbol{\omega}}$ representing the Jacobian with respect to $\boldsymbol{\omega}$. $\frac{\partial \ell}{\partial z_i}$ is the derivative of the loss with respect to the score, a feature of the loss function. Further, the gradients $\nabla_{v_i}^T f$ and $\nabla_{\boldsymbol{\omega}} f$ depend only on the function class we are learning. Indeed, for a linear function class, the desired gradient is given by:

$$\nabla_{\boldsymbol{\omega}} L(\boldsymbol{x}, \boldsymbol{y}, f_{\boldsymbol{\omega}}) = \frac{1}{k} \sum_{i=1}^{k} \frac{\partial \ell}{\partial z_i} \left[ \boldsymbol{\omega}^T \mathcal{J}_{\boldsymbol{\omega}}(\text{NE}(\boldsymbol{x}, f_{\boldsymbol{\omega}})_{i:}) + \text{NE}(\boldsymbol{x}, f_{\boldsymbol{\omega}})_{i:} \right] \tag{2}$$

The remaining and crucial term is the Jacobian of the outcome of the PNE optimizer with respect to the weights: $\mathcal{J}_{\boldsymbol{\omega}}(\text{NE}(\boldsymbol{x}, f_{\boldsymbol{\omega}})_{i:})$. The theorem below proves exactly when this Jacobian exists, and explicitly characterizes it. Once again, for ease of exposition we interpret the input and output of the NE function as a vector.

**Theorem 4.** *For* $\text{NE}(\boldsymbol{x}, f_{\boldsymbol{\omega}}) = \arg\max_{\boldsymbol{x}' \in [0,1]^{kd}} \Phi(\boldsymbol{x}', \boldsymbol{x}, f_{\boldsymbol{\omega}})$ *where* $\Phi$ *is strictly concave and smooth, let* $\boldsymbol{\lambda}$ *and* $\boldsymbol{\mu}$ *denote the dual variables corresponding to the upper and lower bound constraints. Then* $\boldsymbol{x}^*(\boldsymbol{\omega}) = NE(\boldsymbol{x}, f_{\boldsymbol{\omega}})$ *is differentiable with respect to* $\boldsymbol{\omega}$ *everywhere except possibly if there exists an* $x_i^*(\boldsymbol{\omega}) \in \{0, 1\}$ *with the corresponding dual variable set to 0; i.e., if there exists* $i$ *such that* $(x_i^*(\boldsymbol{\omega}) = 1 \wedge \lambda_i^* = 0) \vee (x_i^*(\boldsymbol{\omega}) = 0 \wedge \mu_i^* = 0)$.

The proof (deferred to Appendix C) introduces primal and dual slack variables to capture the KKT conditions, which are necessary and sufficient in the optimization above, in an implicit equation. We show that the Jacobian of a slightly modified but representative version of this implicit function has full rank except possibly at the specified points. Aside from these degenerate points, we can use the implicit function theorem to explicitly compute the gradient. With this hand, the learner is free to use standard gradient descent based algorithms to solve the empirical risk minimization problem.

We also note that this result may be insightful for settings even when the potential function may be non-concave. As we have affine constraints, the KKT conditions are still necessary at local optima (Boyd & Vandenberghe, 2004). Since our approach relies on implicitly characterizing the KKT conditions, we can still use this to compute the derivative at local optima, provided we can find them effectively. Game-theoretically, this would correspond to local Nash Equilibrium (Balduzzi et al., 2018).

## 5 MODELS OF EXTERNALITY

The presented results leverage the strict concavity of the potential function to assert that the PNE is unique, efficiently computable, and PAC learning guarantees possible for the learning problem. Section 3 concluded by noting a more general Condition $(1')$ on the cost and externality to ensure this. This allows us to capture possibly non-convex externality; we give two such examples motivated by the settings we highlighted.

*Proportional Externality:* Externality in strategic classification can model an increased risk in detection/audit due to others also manipulating. In our admissions example, the risk of the university (classifier) suspecting something is amiss is much higher when many students wrongfully claim to be top of their class, as opposed to a handful. As such, it is natural that one's externality due to manipulation increases proportional to the extent to which others in the system also manipulate. We express the externality suffered by agent $i$ in such a setting as:

$$t(\boldsymbol{x}_i', \boldsymbol{x}_i, \boldsymbol{x}_j, \boldsymbol{x}_j') = \tfrac{\beta}{k-1} \sum_{\ell=1}^{d} (x_{i\ell}' - x_{i\ell})^2 (x_{j\ell}' - x_{j\ell})^2$$

We scale by $k-1$ since each agent pays a single cost $c(\cdot)$ but suffers externality from $k-1$ agents; this allows $\alpha$ (the scale constant for the cost) and $\beta$ to be on the same scale. Next, observe that for individuals who do not manipulate, the total externality they incur is 0. This is consistent with one interpretation of the university admissions setting: even if many claim to be top of their class, those who truly are, have nothing to fear. We note this externality is pairwise symmetric and we show below that it satisfies the updated condition (proof in Appendix D).

**Proposition 1.** *Under the cost* $c(\boldsymbol{x}, \boldsymbol{x}') = \alpha \|\boldsymbol{x} - \boldsymbol{x}'\|_2^2$, *the cumulative impact of manipulation under the proportional externality model is smooth and strictly convex for* $\beta < \alpha$.

*Congestion Externality:* In many decision-making scenarios, reported features map to real resources and have downstream consequences beyond classification. Externality can thus model an increased cost for manipulated features due to demand from others. Many countries, for example, deploy immigration policies that favour candidates who pledge to settle in under-populated areas (Picot et al., 2023). Naturally, an influx of applicants may report such intentions and initially move to these areas (with many reneging on this soon after and relocating[10]); this can significantly increase housing and living costs for new immigrants in these underpopulated communities. In such cases, individuals suffer from others manipulating even if they are being honest. Externalities of this nature can be modelled as follows:

$$t(\boldsymbol{x}'_i, \boldsymbol{x}_i, \boldsymbol{x}_j, \boldsymbol{x}'_j) = \tfrac{\beta}{k-1} \sum_{\ell=1}^{d} \exp(-(x'_{i\ell} - x'_{j\ell})^2)$$

Under this externality, when reported values (regardless of their veracity) become more similar, indicating usage of common resources, the externality to agents increases, with the exponential function ensuring this is smooth. Again, it is pairwise symmetric and satisfies our updated convexity condition (proof in Appendix D).

**Proposition 2.** *Under cost $c(\boldsymbol{x}, \boldsymbol{x}') = \alpha||\boldsymbol{x} - \boldsymbol{x}'||_2^2$, the cumulative impact of manipulation under the congestion externality model is smooth and strictly convex for $\beta < \frac{\alpha}{\sqrt{2}}$.*

We note that while these functions capture the spirit of each setting, they are by no means definitive. Indeed there may be other representations for these settings that satisfy our desiderata. Choosing the right model for the right context is an important design goal of the decision-maker.

# 6 EXPERIMENTS

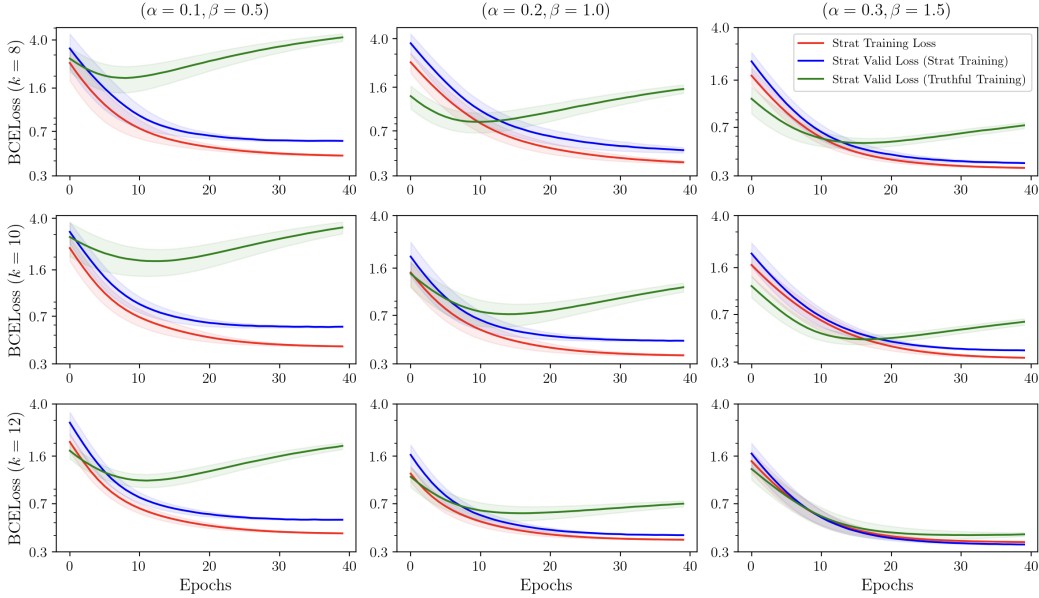

Figure 1: Strategic training (red), corresponding strategic validation losses (blue), and strategic validation loss of a baseline classifier trained with truthful/non-strategic reporting (green). We plot the mean losses over 15 random datasets (with 90% confidence interval) vs training epochs.

We experimentally validate the possibility of training classifiers to be strategy-aware with respect to $k$ agents misreporting to PNE. We use a 2-class synthetic dataset generated by `scikit-learn` and use the `cvxpylayers` library to compute PNE and update gradients (Pedregosa et al., 2011;

---

[10]Canada uses such a policy and reported provincial retention rates as low as 39% (Picot et al., 2023).

Agrawal et al., 2019). We use a linear classifier that returns the probability of assigning the positive class. Agent utilities are computed as per Equation (1), and we use the $\ell_2$ norm square for the cost — $c(\boldsymbol{x}_i, \boldsymbol{x}_i') = \alpha||\boldsymbol{x}_i - \boldsymbol{x}_i'||_2^2$ — and the convex externality discussed in Section 3: $t(\boldsymbol{x}_i, \boldsymbol{x}_i', \boldsymbol{x}_j, \boldsymbol{x}_j') = \beta(||\boldsymbol{x}_i' - \boldsymbol{x}_i||_2 + ||\boldsymbol{x}_j' - \boldsymbol{x}_j||_2)^2$, with $T(\boldsymbol{x}_i, \boldsymbol{x}_i', \boldsymbol{X}, \boldsymbol{X}') = \frac{\beta}{k-1}\sum t(\boldsymbol{x}_i, \boldsymbol{x}_i', \boldsymbol{x}_j, \boldsymbol{x}_j')$; we normalize by $(k-1)$ to make the $\beta$ term interpretable since each agent face externality from $(k-1)$ others. During training, we sample a set of $k$ agents from the training set, compute their PNE under the current classifier, use these PNE features to determine the outcome and loss, and then update gradients. We denote this as *strategic training*, and the corresponding loss curve is in red in figure 2. After each epoch, we run our model on the validation/test dataset, wherein we carry out the same procedure: sample $k$ agents, compute PNE, and then classify. The resulting loss is plotted in blue in Figure 2. We ablate this experiment over different values of $k$ and different strengths of cost and externality.

Given that the training and validation curves are decreasing in lockstep, our experiments indicate that the model is indeed generalizing. Further, the generalization error seems to decrease as the intensity of the cost and externality increase. As a baseline to compare against, we train a truthful/non-strategic classifier to optimality and consider the implications of using this classifier at test-time where agents are strategic (captured in the green curve). This is meant to simulate the deployment of non-strategy-aware classifiers in a strategic setting. We note that for nearly all configurations, this baseline performs poorly, as expected. That said, while it is exceedingly poor for lower intensities of cost and externalities, as the intensity increases, the baseline performance is relatively improved. This trend is somewhat expected since at higher intensities, the negative impact of manipulation is higher, incentivizing many to stay close to their true reports. More interestingly, we note that as $k$ increases, the non-strategic baseline performance also improves. While this is less intuitive, as $k$ increases, any single agent's influence on total externality diminishes, and this quantity becomes an average over a larger set of values. This has a stabilizing effect on the externality an agent experiences which may lead to more moderate equilibrium strategies. Nonetheless, it would be intriguing future work to better understand how aggressively the equilibrium reports shift from true values as $k$ grows larger. In Appendix E, we compare against another baseline where the model is trained only considering cost and not externality, mimicking the classic setup of Hardt et al. (2016).

## 7 DISCUSSION

This paper studies a fundamental question within the strategic classification paradigm: what is the effect of inter-agent externalities on both the agents and the classifier? It is no longer reasonable to assume agents simply best respond to the classifier; rather, the Nash Equilibrium of the induced game becomes the natural solution concept, and we provide a set of conditions whereupon this equilibrium is unique and efficiently computable. The classifier, on the other hand, must learn an optimal model with respect to such an induced equilibrium. We show that this Stackelberg-Nash Equilibrium can be learned in a PAC sense and its loss gradients computable. This paper shows the possibility of deploying loss-minimizing classifiers robust against rich manipulation dynamics.

A limitation of our work is the structural assumptions on externality. Their pairwise nature gives rise to a potential game, and the convexity assumption ensures this is efficiently computable and satisfies Lipschitz regularity conditions. This leads to an intriguing open question: what learning guarantees, if any, can be given for non-concave potential functions where equilibria may not be unique or well-behaved? Can we precisely specify the necessary conditions (our results give a set of sufficient conditions) on externality to ensure both learnability and robustness? Understanding this is both technically and practically interesting. Another important direction is assuming the learner does *not* know the cost and externality model and must learn them by observing equilibrium reports over multiple interactions. This closely parallels the literature on online strategic classification (Dong et al., 2018; Chen et al., 2020) and learning in games (Cesa-Bianchi & Lugosi, 2006). Given the high stakes, characterizing the welfare properties of multi-agent strategic interaction is also imperative. This can involve parametrizing the price of anarchy (Roughgarden, 2010) in terms of a given classifier or simultaneously optimizing for welfare alongside robustness. In settings with transferrable utility, it may also be pertinent to consider coalition dynamics within the strategic model. Lastly, engaging with pertinent stakeholders to accurately capture real costs and externality and experimentally validate our model is necessary. This paper lays the groundwork for these important questions that arise when deploying classifiers in strategic multi-agent settings.

ACKNOWLEDGEMENTS

Ariel Procaccia was partially supported by the National Science Foundation under grants IIS-2147187 and IIS-2229881; by the Office of Naval Research under grants N00014-24-1-2704 and N00014-25-1-2153; and by a grant from the Cooperative AI Foundation. Yiling Chen was partially supported by Amazon and the National Science Foundation under grant IIS-2147187.

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

## A    IMPERFECT INFORMATION SETTING

Our model considers the agent interaction induced by a classifier as a simultaneous game with the Pure Nash Equilibrium (PNE) solution concept. In general, PNE are defined for complete information settings since the agent needs to evaluate their best-response for them to verify an equilibrium. Nonetheless, recent literature on strategic classification literature has considered settings wherein agents have uncertainty about some relevant information (Jagadeesan et al., 2021; Bechavod et al., 2022; Xie & Zhang, 2024).

In this vein, we show an interesting relationship between our complete information PNE and the equilibrium under such uncertainty. More formally, consider a variant of our model where each agent $i$ does not exactly know the true features $X_{-i}$ of the remaining agents, but instead has an estimate $\hat{X}_{-i}$ about the features of the remaining agents, with $X_{-i} = \hat{X}_{-i} + b_i$, with $b_i$ capturing the bias/error of the estimate[11]. Then each agent computes their utility with respect to this possibly biased estimate and evaluates joint strategy based on this. Formally:

$$u_i(\boldsymbol{x}_i, \boldsymbol{x}_i', \hat{X}_{-i}, X_{-i}', f_{\boldsymbol{\omega}}) = f_{\boldsymbol{\omega}}(\boldsymbol{x}_i')g^+(\boldsymbol{x}_i) - c(\boldsymbol{x}_i, \boldsymbol{x}_i') - T(\boldsymbol{x}_i, \boldsymbol{x}_i', \hat{X}_{-i}, X_{-i}')] \tag{3}$$

We note that only the externality term is affected by the uncertainty. We now show that at the complete information PNE $X^q = (\boldsymbol{x}_1^q, \ldots, \boldsymbol{x}_k^q)$ strategy, no agent can improve their biased/imperfect utility by more than $\varepsilon$. Moreover this $\varepsilon$ linearly depends on the bias/error $b$.

**Proposition 3.** *At the complete information PNE strategy $X^q = (\boldsymbol{x}_1^q, \ldots, \boldsymbol{x}_k^q)$, no agent believes they can increase their biased/imperfect utility (equation 3) by more than $\varepsilon = 2\lambda||\boldsymbol{b}_i||$, where $\lambda$ is the Lipschitz constant for the externality function.*

*Proof.* We recall this total externality function $T_i$ is smooth (twice differentiable) and since the domain of features is bounded, this function is also $\lambda$-lipschitz, for some $\lambda \geq 0$. Formally, for any fixed value of $\boldsymbol{x}_i, \boldsymbol{x}_i', X_{-i}'$:

$$|T_i(\boldsymbol{x}_i, \boldsymbol{x}_i', X_{-i}, X_{-i}') - T_i(\boldsymbol{x}_i, \boldsymbol{x}_i', \hat{X}_{-i}, X_{-i}')| \leq \lambda||X_{-i} - \hat{X}_{-i}||_2 \leq \lambda||\boldsymbol{b}_i||_2 \tag{4}$$

Next, suppose that $X^q = (\boldsymbol{x}_1^q, \ldots, \boldsymbol{x}_k^q)$ is the complete information PNE strategy. Then by definition, the following holds for any $\boldsymbol{x}_i'$:

$$f_{\boldsymbol{\omega}}(\boldsymbol{x}_i^q)g^+(\boldsymbol{x}_i) - c(\boldsymbol{x}_i, \boldsymbol{x}_i^q) - T(\boldsymbol{x}_i, \boldsymbol{x}_i^q, X_{-i}, X_{-i}^q) \geq f_{\boldsymbol{\omega}}(\boldsymbol{x}_i')g^+(\boldsymbol{x}_i) - c(\boldsymbol{x}_i, \boldsymbol{x}_i') - T(\boldsymbol{x}_i, \boldsymbol{x}_i', X_{-i}, X_{-i}^q)$$

Using the relations established in equations 4, we have that:

$$f_{\boldsymbol{\omega}}(\boldsymbol{x}_i^q)g^+(\boldsymbol{x}_i) - c(\boldsymbol{x}_i, \boldsymbol{x}_i^q) - T(\boldsymbol{x}_i, \boldsymbol{x}_i^q, \hat{X}_{-i}, X_{-i}^q)] + \lambda||\boldsymbol{b}_i||_2 \geq$$
$$f_{\boldsymbol{\omega}}(\boldsymbol{x}_i')g^+(\boldsymbol{x}_i) - c(\boldsymbol{x}_i, \boldsymbol{x}_i') - T(\boldsymbol{x}_i, \boldsymbol{x}_i', \hat{X}_{-i}, X_{-i}')] - \lambda||\boldsymbol{b}_i||_2$$

The inequality above implies that at a PNE, agent $i$ cannot gain by more than $2\lambda||\boldsymbol{b}_i||_2$ with respect to their biased/imperfect utility function. $\qquad\square$

---

[11]For simplicity, it is easier to think of $\hat{X}_{-i}$ and $X_{-i}$ as vectors of size $(k-1)d$.

## B  Section 3 Proofs

**Proof of Theorem 1**

*Proof.* A potential function encapsulates the change in utility to any agent $i$ due to their unilateral deviation. For us to claim that the proposed function $\Phi$ as a potential function, the following must hold:

$$\forall i : u_i(\boldsymbol{X}, [\boldsymbol{X}'_{-i}; \boldsymbol{x}'_i], f_{\boldsymbol{\omega}}) - u_i(\boldsymbol{X}, [\boldsymbol{X}'_{-i}; \boldsymbol{x}''_i], f_{\boldsymbol{\omega}}) = \Phi(\boldsymbol{X}, [\boldsymbol{X}'_{-i}; \boldsymbol{x}'_i], f_{\boldsymbol{\omega}}) - \Phi(\boldsymbol{X}, [\boldsymbol{X}'_{-i}; \boldsymbol{x}''_i], f_{\boldsymbol{\omega}})$$

Observe that the expression on the left is given by:

$$[f_{\boldsymbol{\omega}}(\boldsymbol{x}'_i) - f_{\boldsymbol{\omega}}(\boldsymbol{x}''_i)]g^+(\boldsymbol{x}_i) - [c(\boldsymbol{x}'_i, \boldsymbol{x}_i) - c(\boldsymbol{x}''_i, \boldsymbol{x}_i)] - \sum_{j \neq i} t(\boldsymbol{x}_i, \boldsymbol{x}'_i, \boldsymbol{x}_j, \boldsymbol{x}'_j) - t(\boldsymbol{x}_i, \boldsymbol{x}''_i, \boldsymbol{x}_j, \boldsymbol{x}'_j)$$

We claim this is equivalent to $\Phi(\boldsymbol{X}, [\boldsymbol{X}'_{-i}; \boldsymbol{x}'_i], f_{\boldsymbol{\omega}}) - \Phi(\boldsymbol{X}, [\boldsymbol{X}'_{-i}; \boldsymbol{x}''_i], f_{\boldsymbol{\omega}})$. Since only $\boldsymbol{x}'_i$ changes to $\boldsymbol{x}''_i$, terms involving gain and cost in the potential function difference match the above. Similarly, all externalities terms not involving $\boldsymbol{x}'_i$ remain the same and only the externalities involving agent $i$ remain. There are exactly $k-1$ such terms and due to symmetry, this is equivalent to $\sum_{j \neq i} t(\boldsymbol{x}_i, \boldsymbol{x}'_i, \boldsymbol{x}_j, \boldsymbol{x}'_j) - t(\boldsymbol{x}_i, \boldsymbol{x}''_i, \boldsymbol{x}_j, \boldsymbol{x}'_j)$. □

**A note on global externality:**

There are scenarios wherein the externality suffered by an agent is global and cannot be factorized in a pairwise fashion. For a set of reported values $\boldsymbol{X}' = (\boldsymbol{x}'_1, \ldots, \boldsymbol{x}_k)$, let $T(\boldsymbol{X}, \boldsymbol{X}')$ denote the externality suffered by any agent due to the global/total set of decisions of everyone. This sort of externality is spiritually akin to a *tragedy of the commons* scenario (common in modeling pollution, public health, etc) where all agents equally suffer due to the combined actions of the whole (Ostrom, 2008). For such settings, the agent dynamics can be captured by the following potential function:

$$\Phi(\boldsymbol{X}, \boldsymbol{X}', f_{\boldsymbol{\omega}}) = \sum_{i=1}^{k} f_{\boldsymbol{\omega}}(\boldsymbol{x}'_i)g^+(\boldsymbol{x}_i) - \sum_{i=1}^{k} c(\boldsymbol{x}_i, \boldsymbol{x}'_i) - T(\boldsymbol{X}, \boldsymbol{X}'). \tag{5}$$

Indeed, any agent $i$ shifting their reported value from $\boldsymbol{x}'$ to $\boldsymbol{x}''$ observes a change in utility exactly equal to: $[f_{\boldsymbol{\omega}}(\boldsymbol{x}'_i) - f_{\boldsymbol{\omega}}(\boldsymbol{x}_i)]g^+(\boldsymbol{x}_i) - [c(\boldsymbol{x}'_i, \boldsymbol{x}_i) - c(\boldsymbol{x}_i, \boldsymbol{x}_i)] - T(\boldsymbol{X}, [\boldsymbol{X}'_{-i}; \boldsymbol{x}'_i]) - T(\boldsymbol{X}, [\boldsymbol{X}'_{-i}; \boldsymbol{x}''_i])$. It is immediate that this is equal to $\Phi(\boldsymbol{X}, [\boldsymbol{X}'_{-i}; \boldsymbol{x}'_i], f_{\boldsymbol{\omega}}) - \Phi(\boldsymbol{X}, [\boldsymbol{X}'_{-i}; \boldsymbol{x}''_i], f_{\boldsymbol{\omega}})$. As such, insofar as $T(\boldsymbol{X}, \boldsymbol{X}')$ is convex and smooth (the assumptions made at the start of section 3) all results in the paper hold for this global model of externality.

**Proof of Theorem 2**

*Proof.* Due to the pairwise symmetry condition, we know that $\sum_{i=1}^{k} T(\boldsymbol{x}_i, \boldsymbol{x}'_i, \boldsymbol{X}, \boldsymbol{X}') = 2\sum_{i=1}^{k}\sum_{j>i} t(\boldsymbol{x}_i, \boldsymbol{x}'_i, \boldsymbol{x}_j, \boldsymbol{x}'_j)$. Hence if each $T(\boldsymbol{x}_i, \boldsymbol{x}'_i, \boldsymbol{X}, \boldsymbol{X}')$ is convex, then the last term in the potential function, $\sum_{i=1}^{k}\sum_{j>i} t(\boldsymbol{x}_i, \boldsymbol{x}'_i, \boldsymbol{x}_j, \boldsymbol{x}'_j)$ is also convex. Combined with the $\ell_2$ norm square cost, which is strictly convex, and $f_{\boldsymbol{\omega}}$ which is linear, it immediately follows that that the potential function is strictly concave. The optimization program for the Nash Equilibrium follows immediately from this.

For concave optimization problems, any local optima is a global optima. Suppose by contradiction there are two global optima $\boldsymbol{X}', \boldsymbol{X}''$, with $\Phi(\boldsymbol{X}, \boldsymbol{X}', f_{\boldsymbol{\omega}}) = \Phi(\boldsymbol{X}, \boldsymbol{X}'', f_{\boldsymbol{\omega}}) = m$. Then strict concavity implies: $\Phi(\boldsymbol{X}, \lambda\boldsymbol{X}' + (1-\lambda)\boldsymbol{X}'', f_{\boldsymbol{\omega}}) > \lambda\Phi(\boldsymbol{X}, \boldsymbol{X}', f_{\boldsymbol{\omega}}) + (1-\lambda)\Phi(\boldsymbol{X}, \boldsymbol{X}'', f_{\boldsymbol{\omega}}) = m$. Since the feasible region is convex, the points on the $\lambda\boldsymbol{X}' + (1-\lambda)\boldsymbol{X}''$ are feasible and thus neither $\boldsymbol{X}'$ or $\boldsymbol{X}''$ could be maximizers. □

# C    SECTION 4 PROOFS

**Proof of Lemma 1**

*Proof.* We begin by proving the Lipschitz continuity of the maximizer of the following generic optimization problem:

$$\boldsymbol{x}^*(\boldsymbol{\omega}) = \arg\max_{\boldsymbol{x}' \in \mathcal{X}} \Psi(\boldsymbol{x}', \boldsymbol{\omega}) \tag{6}$$

where $\Psi$ is strictly concave in $\boldsymbol{x}'$ for any $\boldsymbol{\omega} \in \Omega$ and $\mathcal{X}$ is convex. We know there is always a unique maximizer, and thus the optimal value $\boldsymbol{x}^*(\boldsymbol{\omega})$ is a well-defined function. We also note that our optimization problem satisfies the conditions of Berge's Maximum Theorem (Berge, 1963). Thus, we can immediately conclude that the correspondence from any $\boldsymbol{\omega}$ to the set of maximizers is upper-hemicontinuous. Since our maximizer is unique - i.e. the set is a singleton - it suffices to observe that any set-valued map that is a singleton is upper-hemicontinous if and only if the corresponding function is continuous.

Let $B_\varepsilon(\boldsymbol{\omega})$ refers to an open ball of radium $\varepsilon$ centered at $\boldsymbol{\omega}$. For Lipschitz-ness, we first note that since $\Omega$ is bounded and compact, it suffices to prove this locally. That is, we wish to show that for any $\boldsymbol{\omega}$, there exists constants $L, \varepsilon_{\boldsymbol{\omega}} > 0$ such that for all $\overline{\boldsymbol{\omega}} \in B_{\varepsilon_{\boldsymbol{\omega}}}, ||\boldsymbol{x}^*(\overline{\boldsymbol{\omega}}) - \boldsymbol{x}^*(\boldsymbol{\omega})||_2 \leq L||\overline{\boldsymbol{\omega}} - \boldsymbol{\omega}||_2$. Indeed, this means that for any two $\boldsymbol{\omega}', \boldsymbol{\omega}'' \in \Omega$, we can consider a sequence of distinct point $\boldsymbol{\omega}_1 = \boldsymbol{\omega}', \boldsymbol{\omega}_2, \ldots, \boldsymbol{\omega}_{n-1}, \boldsymbol{\omega}_n = \boldsymbol{\omega}''$ lying on the line $\alpha\boldsymbol{\omega}' + (1-\alpha)\boldsymbol{\omega}''$ such that $||\boldsymbol{\omega}_i - \boldsymbol{\omega}_{i-1}||_2 \leq \varepsilon_{\boldsymbol{\omega}_{i-1}}$. Then we have the following, which means it suffices to focus on local Lipschitz-ness:

$$||\boldsymbol{x}^*(\boldsymbol{\omega}'') - \boldsymbol{x}^*(\boldsymbol{\omega}')||_2 \leq \sum_{i=1}^{n} L||\boldsymbol{\omega}_i - \boldsymbol{\omega}_{i-1}||_2 = L||\boldsymbol{\omega}'' - \boldsymbol{\omega}'||_2$$

We next prove a property that holds for the maximizer of our problem under any $\boldsymbol{\omega}$. Fix a $\boldsymbol{\omega}$, and let $\boldsymbol{x}^*$ be the maximizer. Next, consider a Taylor expansion of $\Psi$ at $\boldsymbol{x}^*$. One formulation of this presented in Chapter 2 of Wright & Recht (2022) states for some $\gamma \in (0, 1)$, we have (the dependence on $\boldsymbol{\omega}$ is dropped for now as it is unchanged):

$$\Psi(\boldsymbol{x}', \cdot) = \Psi(\boldsymbol{x}^*, \cdot) + (\boldsymbol{x}' - \boldsymbol{x}^*)^T \nabla_{\boldsymbol{x}'} \Psi(\boldsymbol{x}^*, \cdot) + \frac{1}{2}(\boldsymbol{x}' - \boldsymbol{x}^*)^T \nabla_{\boldsymbol{x}'}^2 \Psi(\boldsymbol{x}^* + \gamma(\boldsymbol{x}' - \boldsymbol{x}^*), \cdot)(\boldsymbol{x}' - \boldsymbol{x}^*)$$

$$\implies \Psi(\boldsymbol{x}', \cdot) \leq \Psi(\boldsymbol{x}^*, \cdot) + \frac{1}{2}(\boldsymbol{x}' - \boldsymbol{x}^*)^T \nabla_{\boldsymbol{x}'}^2 \Psi(\boldsymbol{x}^* + \gamma(\boldsymbol{x}' - \boldsymbol{x}^*), \cdot)(\boldsymbol{x}' - \boldsymbol{x}^*)$$

where the second line follows from Lemma 2.7 in Still (2018), which states that at for any $\boldsymbol{x}' \in \mathcal{X}$, $\nabla_{\boldsymbol{x}'} \Psi(\boldsymbol{x}^*, \boldsymbol{\omega}) \cdot (\boldsymbol{x}' - \boldsymbol{x}^*) \leq 0^{12}$. We also note that $\nabla_{\boldsymbol{x}'}^2 \Psi(\boldsymbol{x}', \cdot)$ is the Hessian matrix which is (1) always symmetric and (2) consists of all strictly negative eigenvalues since $\Psi(\boldsymbol{x}', \cdot)$ is always strictly concave. Since any symmetric matrix can be diagonalized as $Q\Lambda Q^T$, where $Q$ is the matrix of orthonormal eigenvectors and the $\Lambda$ the eigenvalues, we have that (define $\boldsymbol{p} = \boldsymbol{x}' - \boldsymbol{x}^*$):

$$(\boldsymbol{x}' - \boldsymbol{x}^*)^T \nabla_{\boldsymbol{x}}^2 \Psi(\boldsymbol{x}^* + \gamma(\boldsymbol{x}' - \boldsymbol{x}^*), \cdot)(\boldsymbol{x}' - \boldsymbol{x}^*) = \boldsymbol{p}^T Q\Lambda Q^T \boldsymbol{p} = \sum_{i=1}^{n} \lambda_i (\boldsymbol{v}_i^T \boldsymbol{p})^2$$

where $\boldsymbol{v}_i$ is the $i^{th}$ eigenvector. Since we have strictly negative eigenvalues and $Q$ is an orthonormal matrix and thus does not affect the norm of vector it matrix multiplies, we have (where the $\lambda_{min}(\nabla_{\boldsymbol{x}'}^2 \Psi)$ returns the minimum eigenvalue of the Hessian across $\boldsymbol{\omega} \in \Omega$ and $\boldsymbol{x}' \in \mathcal{X}$):

$$(\boldsymbol{x}' - \boldsymbol{x}^*)^T \nabla_{\boldsymbol{x}'}^2 \Psi(\boldsymbol{x}^* + \gamma(\boldsymbol{x}' - \boldsymbol{x}^*), \cdot)(\boldsymbol{x}' - \boldsymbol{x}^*) \leq \lambda_{min}(\nabla_{\boldsymbol{x}'}^2 \Psi)||Q^T \boldsymbol{p}||_2^2 = \lambda_{min}(\nabla_{\boldsymbol{x}'}^2 \Psi)||\boldsymbol{p}||_2^2$$

$$\implies \Psi(\boldsymbol{x}', \cdot) \leq \Psi(\boldsymbol{x}^*, \cdot) + \frac{1}{2}\lambda_{min}(\nabla_{\boldsymbol{x}'}^2 \Psi)||\boldsymbol{x}' - \boldsymbol{x}^*||_2^2$$

Since $\lambda_{min}$ must be strictly negative, it is evident that there is a strictly positive constant $c$, namely $c = \frac{1}{2}\lambda_{min}(\nabla_{\boldsymbol{x}'}^2 \Psi) = \frac{1}{2}\min_{\boldsymbol{x}', \boldsymbol{\omega}}(\lambda(\nabla_{\boldsymbol{x}'}^2 \Psi(\boldsymbol{x}', \boldsymbol{\omega})))$ such that:

$$\Psi(\boldsymbol{x}^*, \boldsymbol{\omega}) - \Psi(\boldsymbol{x}', \boldsymbol{\omega}) \geq c||\boldsymbol{x}' - \boldsymbol{x}^*||_2^2$$

We now turn back to local Lipschitz-ness. Fix a $\overline{\boldsymbol{\omega}} \in \Omega$ and denote the unique maximizer by $\boldsymbol{x}^*(\overline{\boldsymbol{\omega}}) = \overline{\boldsymbol{x}}$. From the relation derived above (Equation C), there exist constants $\delta_1, c > 0$, such that:

$$\Psi(\overline{\boldsymbol{x}}, \overline{\boldsymbol{\omega}}) - \Psi(\boldsymbol{x}', \overline{\boldsymbol{\omega}}) \geq c||\overline{\boldsymbol{x}} - \boldsymbol{x}'||_2^2 \quad \forall \boldsymbol{x} \in \mathcal{X}, ||\boldsymbol{x} - \overline{\boldsymbol{x}}||_2 \leq \delta_1 \tag{7}$$

---

[12]Technically the lemma is for convex functions with a $\geq$, but it can be easily massaged for concave functions

Let $\Psi^* = \Psi(\overline{\boldsymbol{\omega}}, \overline{\boldsymbol{x}})$ denote the optimal value at $\overline{\boldsymbol{\omega}}$. For $\delta$ such that $0 < \delta < \delta_1$, let $2\alpha = \min_{\boldsymbol{x}' \text{ s.t.} ||\overline{\boldsymbol{x}} - \boldsymbol{x}'|| = \delta} \Psi(\overline{\boldsymbol{x}}, \overline{\boldsymbol{\omega}}) - \Psi(\boldsymbol{x}', \overline{\boldsymbol{\omega}})$. Since $\overline{\boldsymbol{x}}$ is the global and strict optimizer at $\overline{\boldsymbol{\omega}}$, it holds that $2\alpha > 0$. Therefore, we have that:

$$\Psi(\boldsymbol{x}', \overline{\boldsymbol{\omega}}) \leq \Psi^* - 2\alpha \quad \forall \boldsymbol{x}' \text{ such that } ||\boldsymbol{x}' - \overline{\boldsymbol{x}}||_2 = \delta \tag{8}$$

We now leverage the continuity of $\Psi()$ with respect to $\boldsymbol{\omega}$ at $(\overline{\boldsymbol{x}}, \overline{\boldsymbol{\omega}})$. For a chosen value $\frac{\alpha}{2}$, there must exist an $\varepsilon_1$ such that for all $\boldsymbol{\omega} \in B_{\varepsilon_1}(\overline{\boldsymbol{\omega}})$, we have that $|\Psi(\overline{\boldsymbol{x}}, \overline{\boldsymbol{\omega}}) - \Psi(\overline{\boldsymbol{x}}, \boldsymbol{\omega})| < \frac{\alpha}{2}$. Similarly, the continuity of $\Psi()$ with respect to $\boldsymbol{\omega}$ at $(\overline{\boldsymbol{\omega}}, \boldsymbol{x}')$ such that $||\boldsymbol{x}' - \overline{\boldsymbol{x}}||_2 = \delta$, implies that for any such $\boldsymbol{x}'$ and chosen value $\frac{\alpha}{2}$, there exists $\varepsilon_{\boldsymbol{x}'}$ such that for all $\boldsymbol{\omega} \in B_{\varepsilon_{\boldsymbol{x}'}}(\overline{\boldsymbol{\omega}})$, we have that $|\Psi(\boldsymbol{x}', \overline{\boldsymbol{\omega}}) - \Psi(\boldsymbol{x}', \boldsymbol{\omega})| < \frac{\alpha}{2}$. Letting $\varepsilon = \min(\varepsilon_1, \min_{\boldsymbol{x}'|\delta = ||\boldsymbol{x}' - \overline{\boldsymbol{x}}||} \varepsilon_{\boldsymbol{x}'})$, the following holds:

$$\forall \boldsymbol{\omega} \in B_\varepsilon(\overline{\boldsymbol{\omega}}), \, \overline{\boldsymbol{x}} : \; \Psi(\overline{\boldsymbol{x}}, \boldsymbol{\omega}) \in \left(\Psi^* - \frac{\alpha}{2}, \Psi^* + \frac{\alpha}{2}\right) \implies \Psi(\overline{\boldsymbol{x}}, \boldsymbol{\omega}) \geq \Psi^* - \frac{\alpha}{2} \tag{9}$$

$$\forall \boldsymbol{\omega} \in B_\varepsilon(\overline{\boldsymbol{\omega}}), \, \boldsymbol{x}' \text{ s.t } ||\boldsymbol{x}' - \overline{\boldsymbol{x}}|| = \delta : \; \Psi(\boldsymbol{x}', \boldsymbol{\omega}) \in \left(\Psi(\boldsymbol{x}', \overline{\boldsymbol{\omega}}) - \frac{\alpha}{2}, \Psi(\boldsymbol{x}', \overline{\boldsymbol{\omega}}) + \frac{\alpha}{2}\right) \tag{10}$$

Combining Equations 8 and 10 we have that for any $\boldsymbol{\omega} \in B_\varepsilon(\overline{\boldsymbol{\omega}})$ and any $\boldsymbol{x}'$ such that $||\boldsymbol{x}' - \overline{\boldsymbol{x}}||_2 \leq \delta$:

$$\Psi(\boldsymbol{x}', \boldsymbol{\omega}) < \Psi(\boldsymbol{x}', \overline{\boldsymbol{\omega}}) + \frac{\alpha}{2} \leq \Psi^* - 2\alpha + \frac{\alpha}{2} \leq \Psi^* - \alpha \tag{11}$$

Fix a $\boldsymbol{\omega}_0 \in B_\varepsilon(\overline{\boldsymbol{\omega}})$. We aim to show Lipschitz continuity between $\overline{\boldsymbol{\omega}}$ and $\boldsymbol{\omega}_0$, and follow a similar structure as Theorem 6.2a in Still (2018). We know from Equation 9 that $\Psi(\overline{\boldsymbol{x}}, \boldsymbol{\omega}_0) \geq \Psi^* - \frac{\alpha}{2}$. We also know from Equation 11 that for any $\boldsymbol{x}'$ on the boundary of the $B_\delta(\overline{\boldsymbol{x}})$ ball, $\Psi(\boldsymbol{x}', \boldsymbol{\omega}_0) \leq \Psi^* - \alpha$. Since $\Psi()$ is concave in $\boldsymbol{x}'$, this implies that the $c = \Psi^* - \frac{\alpha}{2}$ super-level set of $\Psi(\boldsymbol{x}', \boldsymbol{\omega}_0)$ (for a fixed $\boldsymbol{\omega}_0$) must lie within this $B_\delta(\overline{\boldsymbol{x}})$ ball. In other words, the maximizer at $\boldsymbol{\omega}_0$, $\boldsymbol{x}^*(\boldsymbol{\omega}_0) \triangleq \boldsymbol{x}_0$ satisfies $||\boldsymbol{x}_0 - \overline{\boldsymbol{x}}||_2 \leq \delta$. Noting that $\Psi(\overline{\boldsymbol{x}}, \boldsymbol{\omega}_0) - \Psi(\boldsymbol{x}_0, \boldsymbol{\omega}_0) < 0$, we have that:

$$\Psi(\overline{\boldsymbol{x}}, \overline{\boldsymbol{\omega}}) - \Psi(\boldsymbol{x}_0, \overline{\boldsymbol{\omega}}) = [\Psi(\overline{\boldsymbol{x}}, \overline{\boldsymbol{\omega}}) - \Psi(\overline{\boldsymbol{x}}, \boldsymbol{\omega}_0)] - [\Psi(\boldsymbol{x}_0, \overline{\boldsymbol{\omega}}) - \Psi(\boldsymbol{x}_0, \boldsymbol{\omega}_0)] + [\Psi(\overline{\boldsymbol{x}}, \boldsymbol{\omega}_0) - \Psi(\boldsymbol{x}_0, \boldsymbol{\omega}_0)]$$
$$\leq \underbrace{[\Psi(\overline{\boldsymbol{x}}, \overline{\boldsymbol{\omega}}) - \Psi(\overline{\boldsymbol{x}}, \boldsymbol{\omega}_0)]}_{A} - \underbrace{[\Psi(\boldsymbol{x}_0, \overline{\boldsymbol{\omega}}) - \Psi(\boldsymbol{x}_0, \boldsymbol{\omega}_0)]}_{B}$$

Consider next the function $\Psi(\boldsymbol{x}', \overline{\boldsymbol{\omega}}) - \Psi(\boldsymbol{x}', \boldsymbol{\omega}_0) \triangleq g(\boldsymbol{x}')$. Consider $g(\boldsymbol{x}')$ between $\overline{\boldsymbol{x}}$ and $\boldsymbol{x}_0$. Indeed $g(\overline{\boldsymbol{x}}) - g(\boldsymbol{x}_0) = A - B \geq \Psi(\overline{\boldsymbol{x}}, \overline{\boldsymbol{\omega}}) - \Psi(\boldsymbol{x}_0, \overline{\boldsymbol{\omega}})$. By the mean value theorem, there exists a $\alpha \in [0, 1]$ such that:

$$\Psi(\overline{\boldsymbol{x}}, \overline{\boldsymbol{\omega}}) - \Psi(\boldsymbol{x}_0, \overline{\boldsymbol{\omega}}) \leq g(\overline{\boldsymbol{x}}) - g(\boldsymbol{x}_0) = \nabla_{\boldsymbol{x}}[\Psi(\overline{\boldsymbol{x}} + \alpha(\boldsymbol{x}_0 - \overline{\boldsymbol{x}}), \overline{\boldsymbol{\omega}}) - \Psi(\overline{\boldsymbol{x}} + \alpha(\boldsymbol{x}_0 - \overline{\boldsymbol{x}}), \boldsymbol{\omega}_0)] \cdot (\overline{\boldsymbol{x}} - \boldsymbol{x}_0)$$

Recall that for a differentiable function $f(\boldsymbol{\omega})$, we can approximate it near a value $\overline{\boldsymbol{\omega}}$ using the gradient: $f(\boldsymbol{\omega}) = f(\overline{\boldsymbol{\omega}}) + \nabla_{\boldsymbol{\omega}} f(\overline{\boldsymbol{\omega}}) \cdot (\overline{\boldsymbol{\omega}} - \boldsymbol{\omega}) + o(||\overline{\boldsymbol{\omega}} - \boldsymbol{\omega}||)$. So for any $\boldsymbol{x}' \in cl(B_\delta(\overline{\boldsymbol{x}}))$ ($cl$ denotes the closure), define $h(\boldsymbol{\omega}) = \nabla_{\boldsymbol{x}}\Psi(\boldsymbol{x}', \boldsymbol{\omega})$. Thus, we have that:

$$h(\boldsymbol{\omega}_0) = h(\overline{\boldsymbol{\omega}}) + \nabla_{\boldsymbol{\omega}} h(\overline{\boldsymbol{\omega}})(\overline{\boldsymbol{\omega}} - \boldsymbol{\omega}_0) + o(\mathbf{1}||\boldsymbol{\omega}_0 - \overline{\boldsymbol{\omega}}||_2)$$
$$\implies \nabla_{\boldsymbol{x}'}\Psi(\boldsymbol{x}', \boldsymbol{\omega}_0) = \nabla_{\boldsymbol{x}'}\Psi(\boldsymbol{x}', \overline{\boldsymbol{\omega}}) + \nabla^2_{\boldsymbol{\omega}\boldsymbol{x}'}\Psi(\boldsymbol{x}', \overline{\boldsymbol{\omega}})(\overline{\boldsymbol{\omega}} - \boldsymbol{\omega}_0) + o(\mathbf{1}||\boldsymbol{\omega}_0 - \overline{\boldsymbol{\omega}}||_2)$$
$$\implies \nabla_{\boldsymbol{x}'}[\Psi(\boldsymbol{x}', \overline{\boldsymbol{\omega}}) - \Psi(\boldsymbol{x}', \boldsymbol{\omega}_0)] = \nabla^2_{\boldsymbol{\omega}\boldsymbol{x}'}\Psi(\boldsymbol{x}', \overline{\boldsymbol{\omega}})(\overline{\boldsymbol{\omega}} - \boldsymbol{\omega}_0) + o(\mathbf{1}||\boldsymbol{\omega}_0 - \overline{\boldsymbol{\omega}}||_2).$$

Going back to $\Psi(\overline{\boldsymbol{x}}, \overline{\boldsymbol{\omega}}) - \Psi(\boldsymbol{x}_0, \overline{\boldsymbol{\omega}})$ we have the following:

$$\Psi(\overline{\boldsymbol{x}}, \overline{\boldsymbol{\omega}}) - \Psi(\boldsymbol{x}_0, \overline{\boldsymbol{\omega}}) \leq \nabla_{\boldsymbol{x}'}[\Psi(\overline{\boldsymbol{x}} + \alpha(\boldsymbol{x}_0 - \overline{\boldsymbol{x}}), \overline{\boldsymbol{\omega}}) - \Psi(\overline{\boldsymbol{x}} + \alpha(\boldsymbol{x}_0 - \overline{\boldsymbol{x}}), \boldsymbol{\omega}_0)] \cdot (\overline{\boldsymbol{x}} - \boldsymbol{x}_0)$$
$$\leq \max_{\boldsymbol{x}' \in clB_\delta(\overline{\boldsymbol{x}})} \nabla_{\boldsymbol{x}'}[\Psi(\boldsymbol{x}', \overline{\boldsymbol{\omega}}) - \Psi(\boldsymbol{x}', \boldsymbol{\omega}_0)] \cdot (\overline{\boldsymbol{x}} - \boldsymbol{x}_0)$$
$$\leq \max_{\boldsymbol{x}' \in clB_\delta(\overline{\boldsymbol{x}})} (||\nabla^2_{\boldsymbol{\omega}\boldsymbol{x}'}\Psi(\boldsymbol{x}', \overline{\boldsymbol{\omega}})||_2 + 1) \cdot ||\overline{\boldsymbol{\omega}} - \boldsymbol{\omega}_0||_2 \cdot ||\overline{\boldsymbol{x}} - \boldsymbol{x}_0||_2.$$

Let $\gamma = \max_{\boldsymbol{x}', \boldsymbol{\omega}} ||\nabla^2_{\boldsymbol{\omega}\boldsymbol{x}'}\Psi(\boldsymbol{x}', \boldsymbol{\omega})||_2 + 1$. Then using the relation outlined in Equation 7, and noting the fact that $\boldsymbol{x}_0 \in B_\delta(\overline{\boldsymbol{x}})$, we have that:

$$c||\overline{\boldsymbol{x}} - \boldsymbol{x}_0||_2^2 \leq \Psi(\overline{\boldsymbol{x}}, \overline{\boldsymbol{\omega}}) - \Psi(\boldsymbol{x}_0, \overline{\boldsymbol{\omega}}) \leq \gamma||\overline{\boldsymbol{\omega}} - \boldsymbol{\omega}_0||_2 \cdot ||\overline{\boldsymbol{x}} - \boldsymbol{x}_0||_2$$
$$\implies ||\overline{\boldsymbol{x}} - \boldsymbol{x}_0||_2 \leq \frac{\gamma}{c}||\overline{\boldsymbol{\omega}} - \boldsymbol{\omega}_0||_2$$

To relate this to the Nash Equilibrium of an arbitrary $k$ participants, observe that the following is a characterization of the $\text{NE}(\boldsymbol{x}, f_{\boldsymbol{\omega}}, k)$ function (again we treat $\boldsymbol{x}$ as a $\mathbb{R}^{dk_{max}}$ dimensional vector):

$$\text{NE}(\boldsymbol{x}, f_{\omega}, k) = \underset{\boldsymbol{x}' \in [0,1]^{dk_{max}}}{\arg\max} \left[ \Phi(\boldsymbol{x}'_{0:k}, \boldsymbol{x}_{0:k}, f_{\omega}) - \sum_{j=k+1}^{k_{max}} ||\boldsymbol{x}'_j - \boldsymbol{x}_j||_2^2 \right]$$

The first term, the potential function, fits the function signature of $\Psi$ (see the objective in equation 6) since $\boldsymbol{x}$ (the true features) is simply a constant. Second, we note that the optimization problem here is separable since the potential function only uses the first $k$ feature vectors and the sum of norms only involves the remaining $k_{max} - k$ feature vectors. Further, the latter is independent of $\boldsymbol{\omega}$ and will always be set to $\boldsymbol{x}'_j = \boldsymbol{x}_j$ for $k < j \leq k_{max}$ in the maximization program. Thus, we can appeal to the result above for the maximization of $\Phi$ and claim the function $\text{NE}(\boldsymbol{x}, f_{\boldsymbol{\omega}}, k)$ to be $\frac{\gamma}{c}$ lipschitz continuous where $\gamma, c$ are as above but for the largest possible value of $k$, which is $k_{max}$. $\qquad \square$

**Proof of Theorem 3**

*Proof.* Let $\mathcal{S} = \{(k_1, \boldsymbol{X}_1, \boldsymbol{y}_1), \ldots, (k_n, \boldsymbol{X}_n, \boldsymbol{y}_n)\}$ be the training set where $\boldsymbol{X}_i \in \mathbb{R}^{k_{max} \times d}$ and $\boldsymbol{y}_i \in \mathbb{R}^{k_{max}}$. Let $L(\boldsymbol{X}_i, \boldsymbol{y}_i, f_\omega, k_i)$ denote the loss on the $i$-th sample, where $L(\boldsymbol{X}_i, \boldsymbol{y}_i, f_\omega, k_i) = \frac{1}{k_i} \sum_{j=1}^{k_i} \ell(f_\omega(\mathrm{NE}(\boldsymbol{X}_i, f_\omega, k_i)_{j:}), y_{ij})$. Note that the following holds where the expectation is over samples from $\mathfrak{D}$:

$$\hat{R}(f_\omega) = \frac{1}{n} \sum_{i=1}^{n} L(\boldsymbol{X}_i, \boldsymbol{y}_i, f_\omega, k_i) \quad \mathbb{E}[\hat{R}(f_\omega)] = \frac{1}{n} \sum_{i=1}^{n} \mathbb{E}[L(\boldsymbol{X}, \boldsymbol{y}, f_\omega, k)] = R(f_\omega) \quad (12)$$

Our goal is to show the following generalization: $|R(\hat{f}_\omega) - R(f_\omega^*)| \geq \varepsilon$ with low probability. To that end, we use the fact that uniform convergence implies generalization. Formally:

$$R(\hat{f}_\omega) - R(f_\omega^*) \leq |\hat{R}(\hat{f}_\omega) - R(\hat{f}_\omega)| + |\hat{R}(f_\omega^*) - R(f_\omega^*)| \leq 2 \sup_{f_\omega \in \mathcal{F}} |\hat{R}(f_\omega) - R(f_\omega)| \quad (13)$$

where the second inequality follows from the triangle inequality and that $\hat{R}(\hat{f}_\omega) - R(\hat{f}_\omega) \leq 0$. Since the function space is continuous, we will fix a $\zeta$-cover $\mathcal{N}_\omega^\zeta$ of the parameter space $\Omega$. That is, for any $\omega \in \Omega$, there exists a $\omega' \in \mathcal{N}_\omega^\zeta$ such that $||\omega - \omega'|| \leq \zeta$. Since our parameter space is $d$ dimensional ball of norm $r$, it is well known that such a covering can be achieved with $|\mathcal{N}_\omega^\zeta| \leq \left(\frac{2r\sqrt{d}}{\zeta}\right)^d$. For a sample $i$ in our dataset, let $\boldsymbol{z}_i$ denote the vector of scores received. That is, $z_{ij} = \langle \mathrm{NE}(\boldsymbol{X}_i, f_\omega, k_i)_{j:}, \omega \rangle$, and let $z'_{ij}$ denote the score on this sample when classifier $f_{\omega'}$ is used. Since our loss function $\ell$ is $\lambda$-Lipschitz in the score $z_{ij}$, we have that:

$$|L(\boldsymbol{X}_i, \boldsymbol{y}_i, f_\omega, k_i) - L(\boldsymbol{X}_i, \boldsymbol{y}_i, f_{\omega'}, k_i)| = \left| \frac{1}{k_i} \sum_{j=1}^{k_i} \ell(z_{ij}, y_{ij}) - \ell(z'_{ij}, y_{ij}) \right| \leq \frac{\lambda}{k_i} ||\boldsymbol{z_i} - \boldsymbol{z'_i}||_1$$

Next, by making use of the triangle inequality, we have the following:

$$\frac{\lambda}{k_i} ||\boldsymbol{z_i} - \boldsymbol{z'_i}||_1$$

$$= \frac{\lambda}{k_i} \sum_{j=1}^{k_i} |\langle \mathrm{NE}(\boldsymbol{X}_i, f_\omega, k_i)_{j:}, \omega \rangle - \langle \mathrm{NE}(\boldsymbol{X}_i, f_{\omega'}, k_i)_{j:}, \omega' \rangle|$$

$$= \frac{\lambda}{k_i} \sum_{j=1}^{k_i} |\langle \mathrm{NE}(\boldsymbol{X}_i, f_{\omega'}, k_i)_{j:}, \omega \rangle + \langle \mathrm{NE}(\boldsymbol{X}_i, f_\omega, k_i)_{j:} - \mathrm{NE}(\boldsymbol{X}_i, f_{\omega'}, k_i)_{j:}, \omega \rangle - \langle \mathrm{NE}(\boldsymbol{X}_i, f_{\omega'}, k_i)_{j:}, \omega' \rangle|$$

$$\leq \frac{\lambda}{k_i} \sum_{j=1}^{k_i} |\langle \mathrm{NE}(\boldsymbol{X}_i, f_{\omega'}, k_i)_{j:}, (\omega - \omega') \rangle| + |\langle \mathrm{NE}(\boldsymbol{X}_i, f_\omega, k_i)_{j:} - \mathrm{NE}(\boldsymbol{X}_i, f_{\omega'}, k_i)_{j:}, \omega \rangle|$$

$$\leq \frac{\lambda}{k_i} \sum_{j=1}^{k_i} ||\mathrm{NE}(\boldsymbol{X}_i, f_{\omega'}, k_i)_{j:}||_1 \cdot ||(\omega - \omega')||_1 + \frac{\lambda}{k_i} \sum_{j=1}^{k_i} |\langle \mathrm{NE}(\boldsymbol{X}_i, f_\omega, k_i)_{j:} - \mathrm{NE}(\boldsymbol{X}_i, f_{\omega'}, k_i)_{j:}, \omega \rangle|$$

$$\leq \lambda d ||(\omega - \omega')||_1 + \frac{\lambda}{k_i} ||(\mathrm{NE}(\boldsymbol{X}_i, f_\omega, k_i) - \mathrm{NE}(\boldsymbol{X}_i, f_{\omega'}, k_i))\omega||_1$$

where the last inequality follows from the assumption that the feature vectors are bounded in the region $[0, 1]^d$. Next, we make use of the following 3 relations that hold for any matrix $\boldsymbol{V}$ and vector $\omega \in \mathbb{R}^d$: (1) $||\omega||_2 \leq ||\omega||_1 \leq \sqrt{d}||\omega||_2$, (2) submultiplicavity of matrix norms $||V\omega||_2 \leq ||V||||\omega||_2$ and (3) $||V||_2 \leq ||V||_F$, where $||V||_F$ denotes the matrix Frobenius norm:

$$||(\mathrm{NE}(\boldsymbol{X}_i, f_\omega, k_i) - \mathrm{NE}(\boldsymbol{X}_i, f_{\omega'}, k_i))\omega||_1 \leq \sqrt{k_i}||(\mathrm{NE}(\boldsymbol{X}_i, f_\omega, k_i) - \mathrm{NE}(\boldsymbol{X}_i, f_{\omega'}, k_i))\omega||_1$$

$$\leq \sqrt{k_i}||(\mathrm{NE}(\boldsymbol{X}_i, f_\omega, k_i) - \mathrm{NE}(\boldsymbol{X}_i, f_{\omega'}, k_i))||_F ||\omega||_2$$

$$\leq r\sqrt{k_i}||(\mathrm{NE}(\boldsymbol{X}_i, f_\omega, k_i) - \mathrm{NE}(\boldsymbol{X}_i, f_{\omega'}, k_i))||_F$$

Next, we appeal to theorem 1 which establishes the $\eta-$Lipschitz continuity of the NE function in $\boldsymbol{\omega}$ under the $||\cdot||_F$ norm to state the following:

$$r\sqrt{k_i}||(\text{NE}(\boldsymbol{X}_i, f_{\boldsymbol{\omega}}, k_i) - \text{NE}(\boldsymbol{X}_i, f_{\boldsymbol{\omega}'}, k_i))||_F \leq \eta r \sqrt{k_i}||\boldsymbol{\omega} - \boldsymbol{\omega}'||_1$$

Thus in combining the results here, we can state the following, where the last inequality follows from the definition of a $\gamma$ covering:

$$|L(\boldsymbol{X}_i, \boldsymbol{y}_i, f_{\omega}, k_i) - L(\boldsymbol{X}_i, \boldsymbol{y}_i, f_{\omega'}, k_i)| \leq \lambda d||\boldsymbol{\omega} - \boldsymbol{\omega}'||_1 + \frac{\lambda}{k_i} \eta r \sqrt{k_i}||\boldsymbol{\omega} - \boldsymbol{\omega}'||_1 \leq \lambda(d + \eta r)\gamma$$

This essentially bounds the change in empirical and true risk due to using classifier parameters from using parameters in the finite covering $\mathcal{N}_{\boldsymbol{\omega}}^{\zeta}$. Formally, we expand Equation 13 to state the following ($\boldsymbol{\omega}'$ is the closest element in $\mathcal{N}_{\boldsymbol{\omega}}^{\zeta}$ to $\boldsymbol{\omega}$):

$$\sup_{\boldsymbol{\omega} \in \Omega} |R(f_{\boldsymbol{\omega}}) - \hat{R}(f_{\boldsymbol{\omega}})| = \sup_{\boldsymbol{\omega} \in \Omega} |R(f_{\boldsymbol{\omega}'}) - \hat{R}(f_{\boldsymbol{\omega}'}) + R(f_{\boldsymbol{\omega}}) - R(f_{\boldsymbol{\omega}'}) + \hat{R}(f_{\boldsymbol{\omega}'}) - \hat{R}(f_{\boldsymbol{\omega}})|$$

$$\leq \max_{\boldsymbol{\omega}' \in \mathcal{N}_{\boldsymbol{\omega}}^{\zeta}} |R(f_{\boldsymbol{\omega}'}) - \hat{R}(f_{\boldsymbol{\omega}'})| + 2\lambda(d + \eta r)\gamma.$$

By choosing $\zeta = \frac{\varepsilon}{8\lambda(d+\eta r)\gamma}$, we get that

$$\mathbb{P}\left(\sup_{\boldsymbol{\omega} \in \Omega} |R(f_{\boldsymbol{\omega}}) - \hat{R}(f_{\boldsymbol{\omega}})| \geq \frac{\varepsilon}{2}\right) \leq \mathbb{P}\left(\max_{\boldsymbol{\omega}' \in \mathcal{N}_{\boldsymbol{\omega}}^{\zeta}} |R(f_{\boldsymbol{\omega}'}) - \hat{R}(f_{\boldsymbol{\omega}'})| \geq \frac{\varepsilon}{4}\right)$$

Lastly, we can apply Hoeffding's inequality due to Equation 12 and using union bound over $\mathcal{N}_{\omega}^{\delta}$ at the right-hand side of the inequality above, we have that

$$\mathbb{P}\left(\max_{\boldsymbol{\omega}' \in \mathcal{N}_{\boldsymbol{\omega}}^{\zeta}} |R(f_{\boldsymbol{\omega}'}) - \hat{R}(f_{\boldsymbol{\omega}'})|| \geq \frac{\varepsilon}{4}\right) \leq 2|\mathcal{N}_{\boldsymbol{\omega}}^{\zeta}| \exp\left(\frac{-n\varepsilon^2}{8}\right)$$

$$\leq 2\left(\frac{16\lambda(d + \eta r)\gamma}{\varepsilon}\right)^d \exp\left(\frac{-n\varepsilon^2}{8}\right)$$

and the theorem follows by choosing $n$ accordingly. $\qquad\square$

**Proof of Theorem 4**

*Proof.* Note that NE is the solution to a strictly convex optimization problem. Let $z = kd$. We express the Lagrangian of this problem as follows, with the $\boldsymbol{\lambda} \in \mathbb{R}^z$ denoting the dual variables for the upper bound constraint and $\boldsymbol{\mu} \in \mathbb{R}^z$, the duals for the lower bound (recall our feasible region is a box $[0,1]^z$):

$$\mathcal{L}(\boldsymbol{x}', \boldsymbol{\lambda}, \boldsymbol{\mu}, \boldsymbol{\omega}; \boldsymbol{x}) = \Phi(\boldsymbol{x}', \boldsymbol{x}, f_{\boldsymbol{\omega}}) - \sum_{i=1}^{z} \lambda_i(x_i' - 1) + \sum_{i=1}^{z} \mu_i x_i'$$

Since our feasible region is always strictly satisfiable, Slater's condition is always satisfied. When our objective is concave, as modeled throughout the paper, the KKT conditions are both sufficient and necessary for the optimal solution $\boldsymbol{x}^* = \text{NE}(\boldsymbol{x}, \boldsymbol{\omega})$. Since the constraints are affine, in non-concave setting, the KKT conditions are necessary at a local optimum. Define vectors $\boldsymbol{s}_{x+} \in \mathbb{R}^z, \boldsymbol{s}_{x-} \in \mathbb{R}^z, \boldsymbol{s}_{\lambda} \in \mathbb{R}^z, \boldsymbol{s}_{\mu} \in \mathbb{R}^z$ which will be used to denote the primal and dual slacks. We now define the following implicit function $G(\cdot) : \mathbb{R}^{7z \times d} \to \mathbb{R}^{7z}$:

$$G(\boldsymbol{x}', \boldsymbol{\lambda}, \boldsymbol{\mu}, \boldsymbol{s}_{x+}, \boldsymbol{s}_{x-}, \boldsymbol{s}_{\lambda}, \boldsymbol{s}_{\mu}, \boldsymbol{\omega}) = \begin{bmatrix} \nabla_{\boldsymbol{x}'} \mathcal{L}(\boldsymbol{x}', \boldsymbol{\lambda}, \boldsymbol{\mu}, \boldsymbol{\omega}; \boldsymbol{x}) \\ x_i' - 1 + s_{x+,i}^2 \;\; \forall i \in [m] \\ x_i' - s_{x-,i}^2 \;\; \forall i \in [m] \\ \text{diag}(\boldsymbol{\lambda})(1 - \boldsymbol{x}') \\ \text{diag}(\boldsymbol{\mu})(\boldsymbol{x}') \\ \lambda_i - s_{\lambda,i}^2 \;\; \forall i \in [m] \\ \mu_i - s_{\mu,i}^2 \;\; \forall i \in [m] \end{bmatrix}$$

Let $G(\cdot) = \mathbf{0}$. Under this implicit equation, the first row of equations corresponds to the KKT stationarity conditions. The second two rows of equations enforce each $x'_i$ is less than 1 and greater than 0 respectively - the KKT primal feasibility conditions. The fourth and fifth rows of equations correspond to the complementary slack constraints. The last two rows of equations enforce that the dual variables are positive, the KKT dual feasibility condition. Thus, when $G(\cdot) = \mathbf{0}$, it means the KKT conditions are satisfied, and with a concave objective this also implies the solution is optimal. Similarly, for any optimal $\boldsymbol{X}^*, \boldsymbol{\lambda}^*, \boldsymbol{\mu}^*$, since it is feasible, we can always find the corresponding slacks so as to satisfy $G(\cdot) = 0$. Therefore, solutions to this implicit equation fully characterizes the optimal solution.

Let $\boldsymbol{p} = (\boldsymbol{x}', \boldsymbol{\lambda}, \boldsymbol{\mu}, \boldsymbol{s}_x, \boldsymbol{s}_\lambda, \boldsymbol{s}_\mu)$ and we can simplify our equation to $G(\boldsymbol{p}, \boldsymbol{\omega}) = 0$. Any $\boldsymbol{p}$ that satisfies this is the optimal solution for $\boldsymbol{\omega}$. We aim to use the Implicit Function Theorem and to that end, we first compute the Jacobian $\mathcal{J}_{\boldsymbol{z}}(G)$ as follows (the columns represent the derivative with respect to $\boldsymbol{x}', \boldsymbol{\lambda}, \boldsymbol{\mu}, \boldsymbol{s}_{x+}, \boldsymbol{s}_{x-}, \boldsymbol{s}_\lambda, \boldsymbol{s}_\mu$):

$$\mathcal{J}_{\boldsymbol{z}}(G) = \begin{bmatrix} \nabla^2_{\boldsymbol{x}'}\Phi(\boldsymbol{x}', \boldsymbol{x}, f_{\boldsymbol{\omega}}) & -\boldsymbol{I} & \boldsymbol{I} & \boldsymbol{0} & \boldsymbol{0} & \boldsymbol{0} & \boldsymbol{0} \\ \boldsymbol{I} & \boldsymbol{0} & \boldsymbol{0} & \text{diag}(2)\boldsymbol{s}_{x+} & \boldsymbol{0} & \boldsymbol{0} & \boldsymbol{0} \\ \boldsymbol{I} & \boldsymbol{0} & \boldsymbol{0} & \boldsymbol{0} & \text{diag}(-2)\boldsymbol{s}_{x-} & \boldsymbol{0} & \boldsymbol{0} \\ \text{diag}(\boldsymbol{\lambda}) & \boldsymbol{I}\boldsymbol{x} & \boldsymbol{0} & \boldsymbol{0} & \boldsymbol{0} & \boldsymbol{0} & \boldsymbol{0} \\ -\text{diag}(\boldsymbol{\mu}) & \boldsymbol{0} & -\boldsymbol{I}\boldsymbol{x} & \boldsymbol{0} & \boldsymbol{0} & \boldsymbol{0} & \boldsymbol{0} \\ \boldsymbol{0} & \boldsymbol{I} & \boldsymbol{0} & \boldsymbol{0} & \boldsymbol{0} & \text{diag}(-2)\boldsymbol{s}_\lambda & \boldsymbol{0} \\ \boldsymbol{0} & \boldsymbol{0} & \boldsymbol{I} & \boldsymbol{0} & \boldsymbol{0} & \boldsymbol{0} & \text{diag}(-2)\boldsymbol{s}_\mu \end{bmatrix}$$

We note that the first $m$ columns are always linearly independent owing to the block of identity matrices in the second and third rows of block matrices. Similarly, the second and third sets of $m$ columns are linearly independent owing to the identity in the first row. The remaining columns correspond to slack variables, which we now address. A constraint $i$ is active if $x^*_i \in \{0, 1\}$. Under the KKT conditions, the Lagrange multiplier for inactive constraints must be 0. Let $\mathcal{I} \subseteq [2m]$ denote a set of active constraints. Let $\mathcal{I}(\boldsymbol{\omega}) = \{i | x^*_i(\boldsymbol{\omega}) = 0\} \cup \{i + m | x^*_i(\boldsymbol{\omega}) = 1\}$. We define the inverse mapping $\boldsymbol{\omega}(\mathcal{I})$ as follows: $\boldsymbol{\omega}(\mathcal{I}) = \{\boldsymbol{\omega} \in \boldsymbol{\Omega} | I(\boldsymbol{\omega}) = \mathcal{I}\}$. It is immediate that the set $\{\boldsymbol{\omega}(\mathcal{I}) \, \forall \mathcal{I} \in [2m]\}$ is a partition of the parameter space $\boldsymbol{\Omega}$.

Fix an $\mathcal{I}$ and without loss of generality, let $i$ constraints from $\boldsymbol{\lambda}$ and $j$ constraints from $\boldsymbol{\mu}$ be active. Then for $\boldsymbol{\omega} \in \boldsymbol{\omega}(\mathcal{I})$, it suffices to consider $\boldsymbol{\lambda}_i \in \mathbb{R}^i$ and $\boldsymbol{\mu}_j \in \mathbb{R}^j$ - ie the dual variables corresponding to the active constraints - since the others will be 0 under KKT complementary slackness condition. Similarly, we need only consider the slacks for the inactive constraints since those are the only ones free. Thus for $\boldsymbol{\omega} \in \boldsymbol{\omega}(\mathcal{I})$ we can simplify the general function $G$ to a function $G' : \mathbb{R}^{3m+i+j} \times \mathbb{R}^d \to \mathbb{R}^{3m+i+j}$ by focusing only on the inactive primal slacks and the active dual variables and their corresponding slack.

We note that by definition, the primal slack for inactive constraints cannot be 0 by definition (otherwise those constraints would be active). Hence the columns corresponding to those are also independent. Thus, if the dual slacks for the active constraints are non-zero, then the Jacobian for $G'$ has full rank. This allows us to apply the implicit function theorem and state the following:

$$\mathcal{J}_{\boldsymbol{\omega}}(\text{NE}(\boldsymbol{x}, f_{\boldsymbol{\omega}})) \triangleq \frac{\partial \text{NE}(\boldsymbol{x}, f_{\boldsymbol{\omega}})}{\partial \boldsymbol{\omega}} = \mathcal{J}_{\boldsymbol{p}}^{-1}(G'(\boldsymbol{p}, \boldsymbol{\omega}))\mathcal{J}_{\boldsymbol{\omega}}(G'(\boldsymbol{p}, \boldsymbol{\omega}))$$

If however some of these dual slacks are zero, then it suggests that some of these constraints are redundant and the Jacobian of $G'$ may not be invertible. This is formally equivalent to:

$$\exists i \,|\, (x^*_i = 1 \wedge \lambda^*_i = 0) \vee (x^*_i = 0 \wedge \mu^*_i = 0)$$

Indeed, this is a degenerate situation known as strict complementary failure (Nocedal & Wright, 1999) and represents the threshold or boundary of the $w(\mathcal{I})$ region where one set of constraints is becoming active and another inactive. $\qquad\square$

# D   SECTION 5 PROOFS

**Proof of Proposition 1**

*Proof.* We express the cumulative impact of manipulation under the stated conditions as follows:

$$\alpha \sum_{i=1}^{k} \sum_{\ell=1}^{d} (x'_{i\ell} - x_{i\ell})^2 + \tfrac{\beta}{k-1} \sum_{i=1}^{k} \sum_{j>i} \sum_{\ell=1}^{d} (x'_{i\ell} - x_{i\ell})^2 (x'_{j\ell} - x_{j\ell})^2$$

$$= \sum_{\ell=1}^{d} \left[ \alpha \sum_{i=1}^{n} (x'_{i\ell} - x_{i\ell})^2 + \tfrac{\beta}{k-1} \sum_{i=1}^{k} \sum_{j>i} (x'_{i\ell} - x_{i\ell})^2 (x'_{j\ell} - x_{j\ell})^2 \right]$$

Since the sum of strictly convex functions is convex, it suffices to show that each inner summand is strictly convex. For a fixed $\ell$, we observe that since since there are exactly $\frac{k(k-1)}{2}$ pairs satisfying $j > i$, and in each feature $\boldsymbol{x}_i$ appears in exactly $(k-1)$ of these pairs, the inner summand can be written as follows:

$$\forall \ell : \sum_{i=1}^{k} \sum_{j>i} \tfrac{\alpha}{k-1} (x'_{il} - x_{il})^2 + \tfrac{\beta}{k-1} (x'_{i\ell} - x_{i\ell})^2 (x'_{j\ell} - x_{j\ell})^2 + \tfrac{\alpha}{k-1} (x'_{jl} - x_{jl})^2 \tag{14}$$

Again since convexity is preserved in summation, we only need to show strong convexity with respect to the summands, each of whom is a function of two variables ($x'_{i\ell}$ and $x'_{j\ell}$ since $\boldsymbol{x}_i$ and $\boldsymbol{x}_j$ are constants). For an arbitrary summand index by $(i, j)$, let $u_i = (x'_{i\ell} - x_{i\ell})$ and $u_j = (x'_{j\ell} - x_{j\ell})$. Then the Hessian (upon multiplying Equation 14 by $k - 1$, which does not affect convexity), is:

$$\nabla^2_{x'_{i\ell}, x'_{j\ell}} = \begin{bmatrix} 2\alpha + 2\beta u_j^2 & 4\beta u_i u_j \\ 4\beta u_i u_j & 2\alpha + 2\beta u_i^2. \end{bmatrix}$$

The determinant of this Hessian, when simplified, is given by:

$$\det(\nabla^2_{x'_{i\ell}, x'_{j\ell}}) = 4\alpha^2 + 4\alpha\beta u_i^2 + 4\alpha\beta u_j^2 - 12\beta^2 u_i^2 u_j^2$$

Since we wish to show the determinant is strictly positive, $\alpha^2 + \alpha\beta u_i^2 + \alpha\beta u_j^2 > 3\beta^2 u_i^2 u_j^2$. As feature vectors are bounded, $u_i, u_j \in [-1, 1]$, and our condition is $\alpha > \beta$, the following holds:

$$\alpha^2 + \alpha\beta u_i^2 + \alpha\beta u_j^2 > \beta^2 + \beta^2 u_i^2 + \beta^2 u_j^2 \geq \beta^2 u_i^2 u_j^2 + \beta^2 u_i^2 + \beta^2 u_j^2$$
$$\geq \beta^2 u_i^2 u_j^2 + 2\beta^2 u_i^2 u_j^2 = 3\beta^2 u_i^2 u_j^2,$$

where the second last transition uses the fact that $2u_i^2 u_j^2 \leq u_i^2 + u_j^2$ in the feature vector range.   $\square$

**Proof of Proposition 2**

*Proof.* We express the cumulative impact of manipulation under the stated conditions as follows:

$$\alpha \sum_{i=1}^{k} \sum_{\ell=1}^{d} (x'_{i\ell} - x_{i\ell})^2 + \tfrac{\beta}{k-1} \sum_{i=1}^{k} \sum_{j>i} \sum_{\ell=1}^{d} \exp(-(x'_{i\ell} - x'_{j\ell})^2)$$

$$= \sum_{\ell=1}^{d} \left[ \alpha \sum_{i=1}^{n} (x'_{i\ell} - x_{i\ell})^2 + \tfrac{\beta}{k-1} \sum_{i=1}^{k} \sum_{j>i} \exp(-(x'_{i\ell} - x'_{j\ell})^2) \right]$$

Since the sum of strictly convex functions is convex, it suffices to show that each inner summand is strictly convex. For a fixed $\ell$, we observe that since since there are exactly $\frac{k(k-1)}{2}$ pairs satisfying $j > i$, and in each feature $\boldsymbol{x}_i$ appears in exactly $(k-1)$ of these pairs, the inner summand can be written as follows:

$$\forall \ell : \sum_{i=1}^{k} \sum_{j>i} \tfrac{\alpha}{k-1} (x'_{il} - x_{il})^2 + \tfrac{\beta}{k-1} \exp(-(x'_{i\ell} - x'_{j\ell})^2) + \tfrac{\alpha}{k-1} (x'_{jl} - x_{jl})^2. \tag{15}$$

Again, since convexity is preserved in summation, we only need to show strong convexity with respect to the summands, each of whom is a function of two variables ($x'_{i\ell}$ and $x'_{j\ell}$). For an arbitrary summand index by $(i, j)$, let $u = (x'_{i\ell} - x'_{j\ell})$. Then the Hessian (upon multiplying Equation 15 by $k - 1$, which does not affect convexity) is given by:

$$\nabla^2_{x'_{i\ell}, x'_{j\ell}} = \begin{bmatrix} 2\alpha + 2\beta e^{-u^2}[2u^2 - 1] & 2\beta e^{-u^2}[1 - 2u^2] \\ 2\beta e^{-u^2}[1 - 2u^2] & 2\alpha + 2\beta e^{-u^2}[2u^2 + 1]. \end{bmatrix}$$

A positive definite matrix is equivalent to a matrix with all positive principal minors. When $\beta < \frac{\alpha}{\sqrt{2}} < \alpha$, we note the the the $(0, 0)$ index (and thus the first principal minor) is positive. The second principal minor is the determinant, which for our $2 \times 2$ matrix above is given by:

$$\det(\nabla^2_{x'_{i\ell}, x'_{j\ell}}) = 4\alpha^2 + 16\alpha\beta e^{-u^2}u^2 + 4\beta^2 e^{-2u^2}[4u^2 - 2].$$

Note that the middle term is always positive. The last term is the most negative when $u = 0$, which results in it being $-8\beta^2$. It is evident that can be well compensated for by the first term since using our relation between $\beta$ and $\alpha$, we have that: $8\beta^2 < 8\left(\frac{\alpha}{\sqrt{2}}\right)^2 = 4\alpha^2$. $\qquad\square$

## E ADDITIONAL EXPERIMENTS

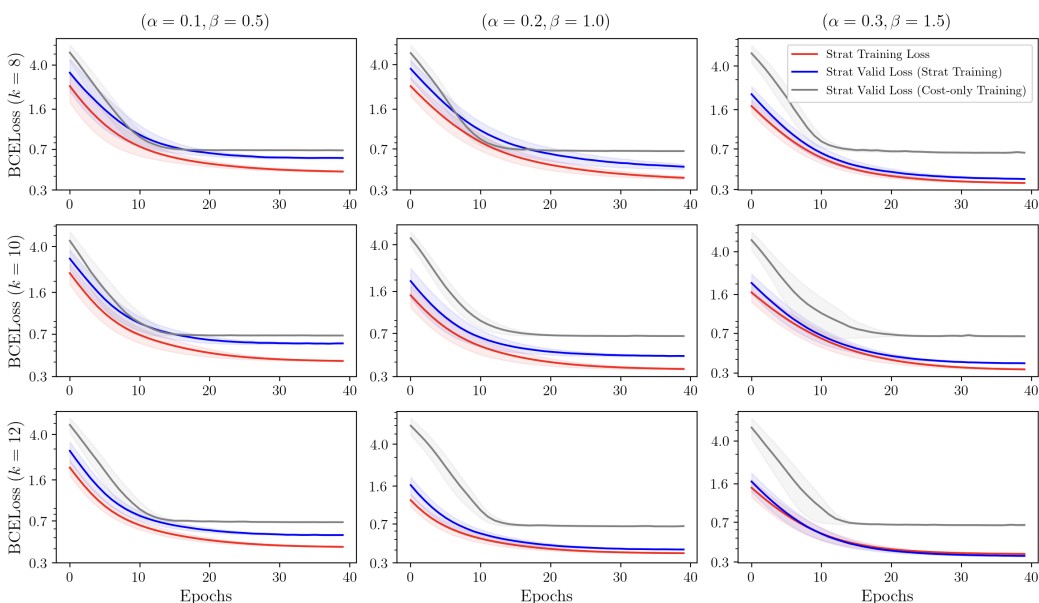

Figure 2: Strategic training (red), corresponding strategic validation losses (blue), and strategic validation loss of a classifier trained only considering cost (gray). We plot the mean losses over 15 random datasets (with 90% confidence interval) vs training epochs.

We also consider comparing our experimental results against a *cost-only* baseline. During the training of this classifier, the agents manipulate their features only considering the cost (and not the externality). In this setting, agents have a dominant strategy since the cost does not depend on the reports of the others. The grey curve plots the strategic validation loss of this classifier, wherein agents manipulate considering both cost and externality and reach a PNE. This baseline is essentially trying to capture the performance of a classifier trained in the classical model of Hardt et al. (2016) (which considers only cost) when deployed in settings with externality.

We first note that the cost-only baseline performs worse than the classifier trained with both cost and externality considerations. This is naturally expected since at validation manipulation occurs considering both. That said, this baseline performs relatively better when the intensity of the cost

and externality are low. This is intuitive since at low intensities, the agents are relatively free to misreport to increase gain, a dynamic captured by both classifiers. As intensities of both cost and externality increase, the cost-only classifier only captures part of the story, ignoring the relatively large negative impacts of manipulation arising from externality. This likely leads the cost-only trained model to anticipate more extreme misreporting than what happens in practice, leading to relatively worse validation performance at these higher intensities. Lastly, we note that $k$ does not seem to have any meaningful effect on the performance of this cost-only baseline.

