# OpenReview forum: "Strategic Classification With Externalities"
_ICLR.cc/2025/Conference — ICLR 2025 Poster_

### Official Review · Reviewer_soro · 2024-10-16

**Soundness:** 4
**Presentation:** 3
**Contribution:** 3
**Rating:** 8
**Confidence:** 3

**Summary:**

This paper serves as the first framework to consider externalities in a multi-agent strategic classification (SC) game. The SC game is still a Stackelberg game with the externality modeled as an extra cost term $T(x_i, x_i', \mathbf{X}, \mathbf{X}')$. Assuming the externalities are pairwise symmetric and *the total cost and the sum of all combinations of externality* is smooth and strictly convex, the authors proved that a unique Nash Equilibrium ($NE$) exists for the post-manipulate features of $k$ agents. Based on this, the authors then characterized the conditions for $NE$ to be Lipschitz in the model parameter and further proved the sample complexity to guarantee the generalization error of the decision policy ($f_\omega$)  in PAC learning. Finally, the authors explored the differentiability of $NE$ w.r.t. model parameter $\omega$, proving the plausibility of using gradient-based algorithms to perform empirical risk minimization.

**Strengths:**

1. Considering externalities in SC settings is well-motivated and the university admission example is quite natural.
2. The paper is well-written and easy to follow. As a purely theoretical work, the authors provided both the PAC learning bound and discussed empirical risk minimization using gradient methods.
3. The technical parts seem to be solid and mostly make sense to me.
4. The paper is situated well in the previous literature. The discussion of [Eilat et al., 2023] is necessary.

**Weaknesses:**

Overall, I do believe the work has some merit and will enrich SC literature. But the paper can still benefit from some more comprehensive discussions.

> Assumptions:

1. the paper assumes a "full-information" SC setting where the decision policy is linear and all strategic agents know the decision policies, cost functions, and externality functions. The authors claimed to adhere to previous literature. However, recent works on SC questioned the plausibility of the assumptions, and a number of them are trying to relax the full information setting (e.g., [1,2,3]). I believe it is at least beneficial to discuss the assumption a bit more.

2. It seems that assuming each agent knows the exact externality $T$ can be more problematic. Each agent has to know the features of all other agents to calculate its utility, and NE is a function of $\mathbf{X}$. Is this realistic in practice and how can each agent know the full $\mathbf{X}$ matrix? At least the authors should make this assumption clear.

3. In Section 5 the authors present some examples of symmetric externality functions, but I still have some confusion. Consider the proportional externality in the admission example. Assume there are 2 agents $i,j$ with $1$-d feature $x_i, x_j$ and they rank $1,2$. Then $j$ seems to be able to manipulate $x_j$ at no externality if $i$ does not manipulate $x_i$.


> Experiments

While most of the theoretical results are clear, I believe the paper can still be improved by adding experiments. It would be great to see the optimal policies under some real datasets (e.g., Give me some credit, ACSIncome) with more practical externality functions. The experiments can demonstrate the sample complexity and the optimization procedure using gradient descent. They can also compare the settings with/without externalities.

[1] Jagadeesan, Meena, Celestine Mendler-Dünner, and Moritz Hardt. "Alternative microfoundations for strategic classification." International Conference on Machine Learning. PMLR, 2021.

[2] Bechavod, Yahav, et al. "Information discrepancy in strategic learning." International Conference on Machine Learning. PMLR, 2022.

[3] Xie, Tian, and Xueru Zhang. "Non-linear Welfare-Aware Strategic Learning." arXiv preprint arXiv:2405.01810 (2024).

**Questions:**

1. Is it reasonable to assume each agent knows the full $\mathbf{X}$?
2. Could you elaborate more on the symmetric externality function in the admission example? Specifically, why the agents who do not manipulate will not impose externality on other agents who manipulate? Am I missing something?
3. Given that you still have one page of space, could you provide at least one set of case studies?

---

> ### Author Response · Authors · 2024-11-20
>
> We thank the reviewer for their helpful suggestions and feedback. We appreciate that they found our problem well-motivated and our technical contributions solid. In the responses, we address their comments in detail, and hope our clarifications will further improve the appreciation of our work.
>
> > It seems that assuming each agent knows the exact externality T can be more problematic. Each agent has to know the features of all other agents to calculate its utility, and NE is a function of X. Is this realistic in practice and how can each agent know the full X matrix?
>
> You are correct that we assume a complete information setting in our model; our solution concept for the classifier induced game is a Pure Nash Equilibrium (PNE) which is only well-defined for complete information settings since the agent needs to evaluate their best-response for them to verify an equilibrium. In settings such as multiple students from a high-school are applying to the same university, complete information may be reasonable.
>
> Nonetheless, you rightly point out that in other scenarios, an agent may not exactly know the features of others (which is needed to compute externality). We can consider then a variant of our model, where each agent $i$ has some estimate $\hat{X_{-i}}$ about the features of the remaining agents, with $X_{-i} = \hat{X_{-i}} + b_i$, with $b_i$ capturing the bias/error of the estimate. Then each agent computes their utility with respect to this possibly biased estimate and evaluates joint strategy based on this. It can be shown that the complete information PNE strategy that we compute is $\varepsilon$ optimal for any agent $i$ under their biased utility estimate, where $\varepsilon$ depends on $b$. In other words, at the complete information PNE strategy, no agent believes they can increase their utility (computed using $\hat{X_{-i}}$) by more than $\varepsilon$.
>
> Appendix A of the updated manuscript contains a formal treatment of this setting, cites the relevant literature mentioned in the review, and formally proves the result above, showing $\varepsilon$ to linearly depend on the bias $b_i$. We think it's an interesting observation, bridging the complete information setting to one where agents have biased/erroneous information; thank you for suggesting this. Future work should further explores this direction of uncertainty and incomplete information; incorporating beliefs and Bayesian perspectives here can also be insightful.
>
> > Consider the proportional externality in the admission example. Assume there are 2 agents i,j with 1-d feature $x_i$,$x_j$ and they rank 1,2. Then $j$ seems to be able to manipulate $x_j$ at no externality if $i$ does not manipulate $x_i$ Could you elaborate more on the externality function in the admission example? Specifically, why the agents who do not manipulate will not impose externality on other agents who manipulate?
>
> In the proportional externality model, it is helpful to think of the cost $c(\cdot)$ as the intrinsic effort/risk in manipulating (i.e. changing your features and the risk of getting caught as part of a routine random audit), and externality $e(\cdot)$ as the extrinsic risk in manipulating (i.e. the principal initiating a widescale audit due to systematic manipulation). A *non*-manipulating agent has nothing to fear on either counts. If an agent does manipulate, their intristic cost/risk is independent of what everyone else does. But their extrinsic cost $e(\cdot)$ depends on what others have done. If very few manipulate, then the principal will not suspect any major loss of integrity, and unlikely to initiate a widespread audit; this implies the manipulating agent suffers minimal externality despite misreporting.
>
> The two-agent case the review pointed out is essentially a limiting case of this phenomena. The agent $i$ who does not manipulate suffers naturally suffers neither cost nor externality. The manipulating agent $j$ still suffers a cost due to manipulating. However, in this model, they suffer no externality since there is no additional extrinsic risk in being caught/audited since the other party was honest.
>
> Lastly, we note that our work is not beholden to this or any specific model of externality. This was an example that we think fits a certain setting. The paper also highlights other externality models (ex: congestion externality) where $j$ could suffer externality even if $i$ is honest.

---

> > ### Author Response · Authors · 2024-11-20
> >
> > > Regarding Experiments
> >
> > We believe our key contribution to be building a model that captures an important but overlooked aspect of strategic machine learning, and settling the associated theoretical questions. That said, we do agree experiments can validating technical aspects of this work and give new insights.
> >
> > Our updated manuscript contains a set of experimental results (see Section 6) on synthetic data that shows the learnability of classifiers that are strategy-aware with respect to cost and externality for a variety of ablations. That is, the loss of this strategic training closely matches the validation loss, wherein agents are also classified based on their PNE outcomes. We also include comparisons against non-strategic baselines. One is trained with truthful data (Figure 1), and another when agents only consider cost (Figure 2). Both are validated against agents who misreport to PNE based on cost and externality. Naturally, these baselines under-perform in validation when compared to the model trained with full strategic awareness. That said, the truthful baseline performs relatively better as the intensity of cost, externality increases (and misreporting becomes expensive) and as the number of agents increase. The cost-only baseline conversely performs more poorly as the intensity increases, likely due to ignoring the relatively large negative impacts of manipulation arising from externality.
> >
> > Beyond technical validation, it is important to also understand the practical implications of using classifiers in real-world multi-agent strategic settings. This is an important question and requires carefully modeling costs and externalities as they arise in each setting, and training the strategy-aware classifiers therein. We believe the theoretical model and technical insights of this paper will facilitate such important research directions.

---

> > > ### Comment · Reviewer_soro · 2024-11-20
> > > **Thanks for your response**
> > >
> > > Thanks for your response and I appreciate the addition of Appendix A and experiments. In my perspective, the paper clears the bar of acceptance.

---

> > > > ### Author Response · Authors · 2024-11-20
> > > >
> > > > Thank you for your time and effort in reviewing the work and our responses!

---

### Official Review · Reviewer_fMjo · 2024-10-31

**Soundness:** 4
**Presentation:** 3
**Contribution:** 4
**Rating:** 8
**Confidence:** 3

**Summary:**

This paper considers agents being classified by a classifier $f_{\omega}$ in a setup where these agents can manipulate their features in order to increase their utility in the classification task. Their utility also depends on the cost they incur to modify their features: the more they modify, the more they pay. Such phenomena naturally arise in tasks such as classifying students for admission at university (you - an agent - would like to manipulate your features to be better classified and get a better university).

The paper brings externalities to the model, which means that one agent's manipulations can affect the outcomes or incentives of others.

The paper frames the interactions as a Stackelberg game, where the principal (the classifier) commits to a strategy first, and is followed by agents responding simultaneously. Beyond theoretical contributions that study the arising equilibria, the paper establishes PAC learning guarantees for classifiers trained in these strategic multi-agent environments. It ends with several considerations on various models of externalities.

**Strengths:**

I think that the problem tackled is very important and relevant. It is nice to see a problem arising from real-world concerns that is studied through a Econ-ML lens.

By combining Stackelberg/Nash game theory concepts with externalities, the authors create a framework that captures both the principal-agent and multi-agent dynamics in a unified model. This approach positions the principal as a leader setting a classifier policy, followed by agents interacting with both the classifier and one another, which is a complex yet analytically tractable setup. Then, based on the first results, they interestingly show that PAC learning is achievable.

The paper is well-written, with a lot of content and interesting theoretical results.

**Weaknesses:**

I think that the assumption of pairwise symmetric externalities is very limiting. Of course, it helps to obtain a model which exhibits nice results and gives Theorem 1 but it seems a bit far from a lot of real-world issues to me. “Equilibrium of Data Markets with Externality” is mentioned as a reference to justify this assumption but does not provide any material to motivate it. “How (Not) to Raise Money”, Goeree et al. is the other reference to justify this assumption, although they only mention symmetry for a very specific case of lottery. I think that more motivation is needed. Removing the assumption would be even better.

More discussion could be provided on the social optimality of the equilibria studied in the paper, for the players as a whole.

**Questions:**

The externality of an agent $j$ suffered by agent $i$ is modeled as $t(x_i, x_i’, x_j, x_j’)$. However, I would intuitively say that externalities are global: the externality suffered by an agent due to a group $J$ might not be factorized as a sum over elements from $J$. Is it possible to have more explanations about your choice?

At the PNE, the agents cannot increase their utility by changing their responses. However, I would be very interested in a discussion about what a coalition of agents could achieve. I feel like a lot of setups involve agents that could behave together and this question is not studied at all in the paper. How would the equilibrium be affected?

“To give an example of a non-linear but convex externality, consider the loan application setting where a bank (the learner) is screening candidates for approval. The more adversely affecting all candidates.” The general setup and this example from your paper in particular remind me the setup of “Strategic Apple Testing”, Harris et al., 2023 (with agents arriving sequentially and manipulating their type). A discussion could be interesting.
The inter-agent externalities explain the whole inter-agent dynamics but I would be interested in a discussion about how the agents could find other ways to interact together, such as through contracting for instance (which is essential in Coase’s theory).

---

> ### Author Response · Authors · 2024-11-20
>
> We thank the reviewer for their insightful comments and questions. We are very glad they found strategic classification in multi-agent settings to be an important problem and commended our framework to capture this. We respond to their comments in detail below and hope it clarifies their questions:
>
> > I would intuitively say that externalities are global: the externality suffered by an agent due to a group $J$ might not be factorized as a sum over elements from $J$.
>
> We agree that there are scenarios wherein the externality suffered by an agent is global and cannot be factorized in a pairwise fashion. Interestingly, our model and all technical results also hold for a general type of global externality. For a set of reported values $X' = (x’_1, \dots, x'_k)$, let $T(X’,X)$ denote externality suffered by any agent due to the global/total set of decisions of everyone. This sort of externality is spiritually akin to a *tragedy of the commons* scenario (common in modeling pollution, public health, etc) where all agents equally suffer due to the combined actions of the whole. The agent-dynamic under such externality can be captured by a potential function, and when $T(X’, X)$ is convex and smooth, our whole suite of technical results on equilibrium and learning hold. We make note of this in the updated manuscript (line 235 and Appendix B) as it is an important flexibility of our work; thank you for mentioning it.
>
> In general, there is a vast literature on modeling and capturing different forms of externality. We posit our work as laying the foundations for studying this phenomena in classification settings and consider it important for future work to extend it to wider class of externalities.
>
> > At the PNE, the agents cannot increase their utility by changing their responses. However, I would be very interested in a discussion about what a coalition of agents could achieve. I feel like a lot of setups involve agents that could behave together and this question is not studied at all in the paper. How would the equilibrium be affected?
>
> We agree that considering coalitions of agents would be a very interesting and non-trivial extension of our model. One possible way to accommodate this is to model agents as having transferrable utilities (e.g. by introducing payments) and consider groups/coalitions of agents, where each group aims to maximize their collective utility. This would still allow for a PNE notion between the groups. However, the dynamic within each group needs to be modeled. Specifically, each group’s collective utility must be apportioned to its members in a fair way that incentivizes them to remain in the group. Concepts like Shapley Value from cooperative game theory will become relevant here.  However, for most strategic classification applications (e.g. college admissions), we think it’s more reasonable to model agents as having non-transferrable utilities. Under a non-transferable utility model, then any stable coalitions have the same action profile as the PNE we characterized in our paper. We discuss these as exciting future directions in the updated manuscript (line 537).
>
> > “Strategic Apple Testing”, (Harris et al., 2023) (with agents arriving sequentially and manipulating their type). A discussion could be interesting. The inter-agent externalities explain the whole inter-agent dynamics but I would be interested in a discussion about how the agents could find other ways to interact together, such as through contracting for instance (which is essential in Coase’s theory)
>
> This hints on some really interesting ideas. Indeed, inter-agent externalities can manifest in different ways. Our externality model intends to capture the externality imposed by the simultaneous actions of agents. It will also be very interesting to consider an online setting, as in the Strategic Apple Testing paper, where a decision maker interacts with agents sequentially. Then, early agents’ actions may impose an externality to a later agent through the change of the classification rule and/or the decision maker’s increasing or decreasing inspection. When agents have transferable utility, then notions like contracting may help restribute the impact of externalities. We hope that our work can inspire followup works on studying externality in strategic ML settings.

---

> ### Comment · Reviewer_fMjo · 2024-11-26
>
> I thank the authors for the discussion which truly helped me to understand the concerns that I had about this paper. I believe that it is a great contribution to the field and I am happy to increase my rating accordingly.

---

### Official Review · Reviewer_T5xz · 2024-11-04

**Soundness:** 3
**Presentation:** 4
**Contribution:** 3
**Rating:** 6
**Confidence:** 5

**Summary:**

The paper introduces a new variant of strategic classification that explicitly models how one agent's feature manipulation can affect other agents' utilities, a form of externalities. The authors model this multi-agent interaction as a Stackelberg game between the classifier (leader) and agents (followers). The paper contains several results under this model:
- Under certain conditions, the agent manipulation game has a unique Pure Nash Equilibrium (PNE)
- The PNE can be efficiently computed under certain technical conditions
- A PAC learning guarantees for learning the optimal classifier
- Some characterization of the optimization problem for learning the classifier
- Two concrete externality models (that can be special cases of their general model): proportional externality and congestion externality.

**Strengths:**

In my opinion, the primary strength of the paper is modeling externality in strategic classification. I believe this aspect of externality has been missing in the long literature of strategic classification.

The authors also did a good job at identifying a clean set of technical conditions (ell_2 norm costs, pairwise symmetric externalities, and convex total externality) for a unique Pure Nash Equilibrium. In particular, the conditions lead to a potential game formulation, enabling the use of well-established game theoretic tools.

These technical conditions also lead to a clean analysis of their PAC learning guarantees. The Lipschitzness condition of NE in Lemma 1 seems quite nice and certainly facilitates the analysis for generalization error.

**Weaknesses:**

The paper's main weakness lies in its treatment of optimization. While the authors elegantly establish the existence and uniqueness of equilibria, they essentially punt on the crucial question of how to actually find the optimal classifier. Their argument - that gradient-based methods often work well for non-convex problems - feels particularly thin in this context. After all, we're not dealing with standard non-convexity here, but rather with a nested optimization where agents are reaching equilibrium inside the classifier's optimization loop.

While the authors derived gradients through the equilibrium, they stopped short of showing these gradients are actually useful. A simple set of experiments on synthetic data or semi-synthetic data would have been useful here. For instance, does gradient descent reliably find good classifiers? How does the convergence behavior change with the number of agents or the strength of externalities? The field of strategic classification has evolved. Recent papers have routinely complemented their theoretical results with experimental validation.

**Questions:**

- Since the authors set up the downstream game as a potential game, perhaps they can make the point that the agents can reach NE through simple best-response dynamics?

- The authors did a good job surveying recent related work on strategic classification, but perhaps missed the following:
** On the Long-term Impact of Algorithmic Decision Policies: Effort Unfairness and Feature Segregation through Social Learning, ICML 2019. This paper models the "information externality" of some agents over others in settings like strategic classification.
** Strategic Instrumental Variable Regression: Recovering Causal Relationships From Strategic Responses, ICML 2022. This paper also accounts for causal effects in strategic classification.

---

> ### Author Response · Authors · 2024-11-20
>
> We thank the reviewer for their constructive comments and feedback. We are glad they found our new model of strategic classification to be clean and filling a gap in the literature. We respond to their comments in detail below.
>
> > A simple set of experiments on synthetic data or semi-synthetic data would have been useful here. For instance, does gradient descent reliably find good classifiers? How does the convergence behavior change with the number of agents or the strength of externalities?
>
> While we consider our key contributions as proposing a new model of strategic classifcation that captures a crucial phenomena and settling the associated theoretical questions, we agree that adding experimental validation would further improve the manuscript. To that end, we have updated the manuscript (see Section 6) with a set of experiments on synthetic data that:
> - Validate the learnability of cost and externality aware classifiers
> - Ablate over the impact of the number of agents and the strength of cost and externality in agent utility
> - Compare against two non-strategy aware baselines.
>
> To breifly summarize the results, we train a linear classifier and compute agent utilities for the corresponding outcome using cost and externality functions discussed in the paper. During training, we sample a set of $k$ agents from the training set, compute their PNE given the current classifier, and use the equilibrium features to compute the classifier outcome and the corresponding loss, and update the classifier with gradient descent algorithms. Our experiments show the loss of this strategic training closely matches the validation loss, wherein agents are also classified based on their PNE outcomes.
>
> We compare this against two baselines. One is where the classifier is trained with truthful data/no strategic considerations (Figure 1) and another where it is trained assuming the agents are strategic but only care about cost (Figure 2). The latter captures existing models of strategic classification. At validation these baseline models classify agents who misreport to PNE based on cost and externality. Naturally, we notice these baselines under-perform in validation when compared to our model above, trained with full strategy awareness. We notice the non-strategic baseline performs relatively better as the intensity of cost, externality increases (and misreporting becomes expensive) and as the number of agents increase. Conversely, the cost-only baseline performs worse as the intensity increases, likely because it overlooks the significant negative impacts of manipulation caused by externalities. This may lead to expecting a more severe manipulation compared to what actually occurs.
>
> In general, the question of practical training and deployment of a model is important; we hope our results lead to more real world empirical analysis of strategic classification with externalities in diverse settings.
>
> > Since the authors set up the downstream game as a potential game, perhaps they can make the point that the agents can reach NE through simple best-response dynamics?
>
> Indeed, the fact that best response dynamics will converge to the Pure Nash Equilibrium is strength of our model. We highlight this in line 252 of the updated manuscript.
>
> > Regarding Related Works
>
> Thank you for pointing out additional relevant literature. We include them in the related works section (Section 1.2) of the updated manuscript.

---

### Official Review · Reviewer_mGwb · 2024-11-06

**Soundness:** 3
**Presentation:** 4
**Contribution:** 3
**Rating:** 6
**Confidence:** 4

**Summary:**

The study a generalization of strategic classification where the game between the principal and agents is Stackelberg, but the game between the agents itself is simultaneous. The principal's goal is to compute the Stackelberg-Nash equilibrium to maximize its utility.

They show:

Under some assumption the PNE of this game is unique and can be computed efficiently.

PAC learning results for the principal.

And they show ERM results through gradient-based methods.

**Strengths:**

I think the writing of the paper is fantastic!
Also, the problem formulation is interesting.

**Weaknesses:**

Besides their novel formulation, their main result is their sample complexity result, but that seems to me to just be using a standard covering argument to bound the discretization error. Do you think you can get tighter bounds using other techniques like Rademacher complexity etc.?

**Questions:**

They assume that the number of agents that participate in the game is not fixed, but comes from the distribution. However, this assumption does not impact the PAC learning result. This assumption is mostly interesting from the modeling point, but are there scenarios where this assumption leads to different outcomes compared to a fixed number of agents?

Line 196, shouldn't it be f_{\omega}(x') instead of x inside the parentheses? If this is correct, then you may want to double-check it throughout the paper, in case it appears in other equations or explanations.

---

> ### Author Response · Authors · 2024-11-20
>
> We thank the reviewer for their insightful comments and commending our problem formulation and model. We address their questions in detail below:
>
> > Do you think you can get tighter bounds using other techniques like Rademacher complexity etc.?
>
> This is an excellent question and we did consider using a Rademacher complexity based approach. Indeed, it is possible to use a symmetrization argument to bound the generalization error in terms of the Rademacher Complexity. In our setting, however, the Rademachar complexity is of the function $f_{\omega}$ applied to the Nash Equilibrium outcome $\text{NE}(X, f_\omega,k)$ (which itself depends on $f_{\omega}$), and not directly on the set of $k$ samples $X$. In other words, the generalization error can be stated in terms of the following quantity:
>
> $$\mathbb{E}[\sup_{\omega\in \Omega} \frac{1}{n} \sum_{i=1}^n \sigma_i \cdot \langle \omega,  NE(X_i, f_\omega, k_i)_j\rangle )]$$
>
> where $\sigma_i$ are Rademacher variables and $\text{NE}(X, f_\omega,k)$  is the $j^{th}$ agent's PNE value. In standard analysis for linear classifiers, the dependence on $\sup_{\omega}$ can be eliminated and Cauchy-Schwartz applied to bound this term. In our case, however, the dependence of $NE(X_i, f_\omega, k_i)$ on $\omega$, mean the supremum cannot be easily elimated to give a meaningful bound. So while the RC approach is feasible, it does not lead to a interpretable bound, leading us to use a covering approach.
>
> > When does the number of agents being dynamic lead to different outcomes vs fixed agents.
>
> This is also an insightful question. When thinking about generalization, each sample in our setting consists of a value $k$ (sampled from a distribution) and the features vectors of $k$ agents. We assume these features vectors are sampled independently from some distribution $D$. While this is reasonable in many of the settings where strategic classification is relevant, one can consider an alternate setting where the $k$ agent feature vectors are sampled from a joint distribution where these feature vectors can be arbitrarily correlated (and not independent). If $k$ is fixed, our results still hold in correlated world. However, if $k$ is random and we are in this correlated world, it is unclear if generalization can hold. In short, for fixed value of $k$, our results hold regardless of correlations within a sample, while for random $k$, we require the feature vectors of the $k$ agents to be independent.

---

> > ### Comment · Reviewer_mGwb · 2024-11-26
> >
> > I thank the authors for their response. At this point, I have no further questions/concerns.

---

### Meta-Review · Area_Chair_5rzd · 2024-12-20

**Metareview:**

This paper examines the problem of strategic classification via a model where agents' feature manipulations can affect others, capturing inter-agent strategic externalities. The authors model the principal-agent interactions as a Stackelberg game and the agents' manipulation dynamics as a simultaneous game. They then show that, under certain assumptions, the pure Nash equilibrium of this agent manipulation game is unique and can be computed efficiently. Building on this, they establish PAC learning guarantees, indicating that classifiers can be learned to minimize loss even when agents manipulate their features to reach a pure Nash equilibrium.

Reviewers commended the paper’s novel approach to incorporating inter-agent externalities in strategic classification, noting its potential to enhance the realism and applicability of models in this domain. The reviewers appreciated in particular the authors' equilibrium uniqueness and computation results. Some concerns were raised regarding the practicality of the proposed methods and the assumptions underlying the authors' model but, for the most part, these were addressed by the authors' rebuttal. In view of all this, the consensus among reviewers was that the paper provides a solid contribution to the problem of strategic classification, and unanimously recommended acceptance.

**Additional Comments On Reviewer Discussion:**

To conform to ICLR policy, I am repeating here the relevant part of the metareview considering the reviewer discussion.

> Reviewers commended the paper’s novel approach to incorporating inter-agent externalities in strategic classification, noting its potential to enhance the realism and applicability of models in this domain. The reviewers appreciated in particular the authors' equilibrium uniqueness and computation results. Some concerns were raised regarding the practicality of the proposed methods and the assumptions underlying the authors' model but, for the most part, these were addressed by the authors' rebuttal. In view of all this, the consensus among reviewers was that the paper provides a solid contribution to the problem of strategic classification, and unanimously recommended acceptance.

---

### Decision · Program_Chairs · 2025-01-22

Accept (Poster)